**Data Availability Statement:** All relevant data are within the paper and its Supporting Information files.

# The metronomic combination of paclitaxel with cholinergic agonists inhibits triple negative breast tumor progression. Participation of M2 receptor subtype

**Alejandro J. Español**[1,2,3☉]*, **Agustina Salem**[1,2☉], **María Di Bari**[3], **Ilaria Cristofaro**[3], **Yamila Sanchez**[1,2], **Ada M. Tata**[3,4]*, **María E. Sales**[1,2]

1 Center of Pharmacological and Botanical Studies (CEFYBO), CONICET, Buenos Aires, Argentine,
2 Department of Pharmacology, School of Medicine, University of Buenos Aires, Buenos Aires, Argentine,
3 Department of Biology and Biotechnologies Charles Darwin, Sapienza University of Rome, Rome, Italy,
4 Center of Neurobiology Daniel Bovet, Sapienza University of Rome, Rome, Italy

☉ These authors contributed equally to this work.
* aespan_1999@yahoo.com (AJE); adamaria.tata@uniroma1.it (AMT)

## Abstract

Triple negative tumors are more aggressive than other breast cancer subtypes and there is a lack of specific therapeutic targets on them. Since muscarinic receptors have been linked to tumor progression, we investigated the effect of metronomic therapy employing a traditional anti-cancer drug, paclitaxel plus muscarinic agonists at low doses on this type of tumor. We observed that MDA-MB231 tumor cells express muscarinic receptors, while they are absent in the non-tumorigenic MCF-10A cell line, which was used as control. The addition of carbachol or arecaidine propargyl ester, a non-selective or a selective subtype 2 muscarinic receptor agonist respectively, plus paclitaxel reduces cell viability involving a down-regulation in the expression of ATP "binding cassette" G2 drug transporter and epidermal growth factor receptor. We also detected an inhibition of tumor cell migration and anti-angiogenic effects produced by those drug combinations *in vitro* and *in vivo* (in NUDE mice) respectively. Our findings provide substantial evidence about subtype 2 muscarinic receptors as therapeutic targets for the treatment of triple negative tumors.

## Introduction

Breast cancer is still the most frequent type of malignancy in women and represents a major and unsolved problem for public health [1, 2]. Luminal and triple negative (TN) represent the two opposite ends of the molecular classification of breast tumors and they thoroughly differ regarding treatment and patients´survival [3]. The TN tumors are typically larger in size, higher grade than other breast cancers, and they also exhibit an aggressive clinical behavior, frequently resulting in early metastatic dissemination, particularly to visceral sites. As a result of these characteristics, TN breast cancers are associated with poor prognosis in comparison to luminal breast tumors [4, 5]. Considering the treatment of TN tumors, classical modalities

**Funding:** AJE; ANPCyT PICT 2015-2017, 2396. Agencia Nacional de Promoción Científica y Tecnológica. The funders had no role in study design, data collection and analysis, decision to publish, or preparation of the manuscript. MES, CONICET PIP 2015-2017, 201501 00239; Consejo Nacional de Investigaciones Científicas y Técnicas; UBA UBACYT 2014-2017, 20020130100168BA Universidad de Buenos Aires. The funders had no role in study design, data collection and analysis, decision to publish, or preparation of the manuscript.

**Competing interests:** The authors have declared that no competing interests exist.

have improved the overall outlook and quality of life for women with this type of breast cancer. However, because of recurrence and/or the development of resistance to cytotoxic drugs administered to patients, produced by a complex mechanism mediated by different types of proteins such as ATP "binding cassette" (ABC) transporters, a considerable amount of patients still succumb to this disease highlighting the need to find new therapeutic approaches [6]. Regarding the latter, the administration of low dose chemotherapy with short drug free intervals, named metronomic therapy emerged as a novel regimen for cancer treatment [7]. It exerts very low incidence of side effects and could add new beneficial actions on immune system and tumor microenvironment [8]. This new strategy also needs the identification of new therapeutic targets to improve the benefits for breast cancer patients.

Non-neuronal cholinergic system (nNCS) has been involved either in physiological or in pathological processes. The nNCS is formed by acetylcholine (ACh), the enzymes that synthesize and degrade ACh and cholinergic receptors expressed in non-neuronal cells. Muscarinic receptors belong to this group of proteins and have been involved in the progression of different type of tumors such as lung, colon and prostate [9–11]. We demonstrated that muscarinic receptors are expressed in tumor samples from patients with breast cancer in different stages and also in human MCF-7 cells derived from a luminal, estrogen-dependent adenocarcinoma, the most frequent type of breast tumor in women [12, 13]. Muscarinic receptors belong to the G-protein coupled receptors family which constitutes the largest family of cell surface receptors involved in signal transduction. Five subtypes have been identified by molecular cloning: M$_1$-M$_5$. Their role in the regulation of important cell functions like mitosis, cell morphology, locomotion and immune response which are key steps during tumor progression has been documented [14, 15]. The long-term activation of these receptors with the agonist carbachol stimulates cytotoxicity either in human or in murine breast tumor cells [16, 17]. In the last years, several reports demonstrated that the activation of subtype 2 muscarinic (M$_2$) receptor subtype by a selective agonist was able to arrest cell proliferation in different tumor cell lines [18, 19]. Moreover, M$_2$ receptor activation reduced cell survival, inducing oxidative stress and severe apoptosis in malignant cells derived from human glioblastoma [20]. MDA-MB231 is a human cell line derived from a TN breast tumor, which does not express estrogen/progesterone receptors or HER2 protein.

The aim of our work is to investigate the ability of a combination of paclitaxel (PX) with a muscarinic agonist both at low doses to inhibit different steps of TN breast tumor progression. In this work, we identified different subtypes of muscarinic receptors in MDA-MB231 cells by Western blot, and demonstrated that the combination of PX plus carbachol or arecaidine propargyl ester (APE), a non-selective or an M$_2$ selective agonist respectively, reduced cell viability, migration, vascular endothelial growth factor-A (VEGF-A) expression as well as *in vivo* angiogenesis. We also observed a down-regulation in the expression of ABCG2 transporter and epidermal growth factor receptor (EGFR) in tumor cells by Western blot, after PX plus carbachol or APE administration revealing that both proteins could be involved in the mechanism of action of this treatment.

## Materials and methods

### Cell culture

The human breast adenocarcinoma cell line MDA-MB231 (CRM-HTB-26) and MDA-MB468 (HTB-132) were acquired from the American Type Culture Collection (ATCC; Manassas, USA) and cultured in DMEM (Invitrogen Inc., Carlsbad, USA) with 2 mM L-glutamine and 80 µg/ml gentamycin, supplemented with 10% heat inactivated FBS (Internegocios SA, Mercedes, Argentine) at 37°C in a humidified 5% CO$_2$ air. The MCF-10A cells (CRL-10317) were

also bought from ATCC and constitute a non-tumorigenic cell line derived from human mammary tissue. These cells were grown on tissue culture plastic dishes in DMEM:F12(1:1) (Invitrogen Inc., Carlsbad, USA) supplemented with 10% FBS, hydrocortisone (0.5 μg/ml), insulin (10 μg/ml), and hEGF (20 ng/ml). Cell lines were detached using the following buffer: 0.25% trypsin and 0.02% EDTA in Ca$^{2+}$ and Mg$^{2+}$ free PBS from confluent monolayers. The medium was changed three times a week. Cell viability was determined by Trypan blue exclusion test and the absence of mycoplasma was observed by Hoechst staining [21].

## Detection of muscarinic receptors by Western blot

Cells (2x10$^6$) were washed twice with PBS and lysed in 1 ml of 50 mM Tris-HCl, 50 mM NaCl, 5 mM NaF, 5 mM MgCl$_2$, 1 mM EDTA, 1 mM EGTA, 5 mM phenylmethanesulfonyl fluoride, 1% Triton X-100 and 10 μg/ml trypsin inhibitor, aprotinin and leupeptin, pH 7.4. Then, lysates were left 1 h in an ice bath, and later centrifuged at 800 g for 20 min at 4˚C. Supernatants were saved at -80˚C and protein concentration was analyzed by the method of Bradford [22]. Samples (80 μg protein per lane) were subjected to 10% SDS-PAGE minigel electrophoresis, transferred to nitrocellulose membranes, and incubated overnight with goat anti-human M$_1$, M$_2$ or M$_3$ receptor polyclonal antibodies or rabbit anti-human M$_4$ or M$_5$ receptor polyclonal antibodies (Santa Cruz Biotechnology Inc., USA) all diluted 1:200. Then strips were incubated with anti-rabbit IgG or anti-goat IgG coupled to horseradish peroxidase, both diluted 1:10000 in 20 mMTris-HCl buffer, 150 mM NaCl and 0.05% Tween 20 (TBS-T) at 37˚C for 1 h. Bands were visualized by chemiluminescence. The results of densitometric analysis were expressed as optical density (O.D.) units relative to the expression of glyceraldehyde 3-phosphate dehydrogenase (GAPDH) (Santa Cruz Biotechnology Inc., USA) [23].

## M$_2$ muscarinic receptor silencing

To silence M$_2$ receptors, tumor cells were transfected with four different small interfering RNA (siRNA) targeting specific sequences of human M$_2$ receptor (ID1129) (Riboxx Life Sciences, Radebeul, Germany). A positive control of transfection Chromo-GAPDH-siRNA (Riboxx Life Sciences, Radebeul, Germany) and a negative control for siRNA assays (NC-siRNA) (SI03650318) (Quiagen, Hilden, Germany) were used. The sequences for M$_2$-siRNAs were the following:

a. (siRNA 1129–1) sense, 5´- AUUUACUACUAAAUCCUCCCCC-3´, antisense 5´-GGGGGGA GGAUUUAGUAGUAAAU-3´;

b. (siRNA 1129–2) sense 5´- AUGUAGCCCAUUUCUUCCCCC-3´, antisense 5´-GGGGGAA GAAAUGGGCUACAU-3´;

c. (siRNA 1129–3) sense 5´-UCCUUUGAGUUUCAGGCUGCCCCC-3´, antisense 5´- GGGG GCAGCCUGAAACUCAAAGGA-3´;

d. (siRNA 1129–4) sense 5´-AGUUACACCUUGACCUAACCCCC-3´, antisense 5´-GGGGGU UAGGUCAAGGUGUAACU-3´.

The cells were seeded in 6-well plates (10$^4$ cells/well) and cultured in 2 ml DMEM with 10% FBS until they were 60–70% confluent. The siRNAs were mixed with Riboxx-Fect according to manufacturer's instructions and then added to wells. The efficiency of the transfection was evaluated by transfecting in separate wells Chromo-GAPDH-siRNA (Riboxx Life Sciences, Radebeul, Germany). The ability of the siRNA pool to down-regulate M$_2$ muscarinic receptor expression was tested 72 h after transfection with 40 nM/well of siRNA and evaluated by Western blot detecting M$_2$ receptor protein expression (23).

## Cell viability assay

The inhibition in cell viability exerted by different treatments for one or three cycles of 48h was analyzed employing the soluble tetrazolium salt 3-(4,5-dimethylthiazol-2-yl)-2,5-diphenyltetrazolium bromide (MTT) colorimetric assay (Life Technologies, Eugene, USA). When cells are alive, MTT is reduced to formazan. $4x10^3$ cells/well were seeded in 96-well plates in culture medium supplemented with 5% FBS and then left to attach overnight. When cells reached 60–70% of confluence they were deprived of FBS 24 h previous to the assay to induce the synchronization of cultures. Then, cells were treated with PX (Bristol-Myers Squibb, Vicente López, Argentine), carbachol or APE (non-selective or selective $M_2$ receptor agonist respectively) alone or in combination in medium supplemented with 2% FBS, during 48 h in triplicate. Also the effect of doxorubicin (Glenmark Generics S.A., Pilar, Argentine) alone or combined with muscarinic agonists was analyzed. To inhibit the action of cholinergic agonists, cells were previously treated with atropine at $10^{-9}$ M or methoctramine at $10^{-5}$ M (non-selective or $M_2$ receptor selective antagonist respectively).

After treatment, to detect viable cells, the medium was replaced by 110 μl of MTT solution that was prepared by diluting 10 μl of 5 mg/ml MTT in PBS, in 100 μl medium free of phenol red and FBS to each well. After incubation for 4 h at 37˚C, the production of formazan was measured by analyzing the absorbance at 540 nm with an ELISA reader (BioTek, Winooski, USA). Values are mean ± S.E.M and results are expressed as the percentage of inhibition in cell viability in comparison to control (cells without treatment).

## ATP binding cassette transporter G2, epidermal growth factor receptor and vascular endothelial growth factor-A detection by Western blot

Cells ($2x10^6$) were treated during three cycles of 48h/each and samples were prepared as it was indicated to detect muscarinic receptors. Then, samples (80 μg protein per lane) were subjected to 8–10% SDS-PAGE minigel electrophoresis, transferred to nitrocellulose membranes, and incubated overnight with a rabbit anti-human ABCG2 polyclonal antibody (Santa Cruz Biotechnology Inc., Dallas, USA) diluted 1:200, a rabbit anti-human EGFR monoclonal antibody (EMD Millipore-MERCK, Temecula, USA) diluted 1:1000 or a rabbit anti-human VEGF-A polyclonal antibody (Abcam, Eugene, USA) diluted 1:1000. Then strips were incubated with horseradish peroxidase-linked anti-rabbit IgG, diluted 1:10000 in TBS-T at 37˚C for 1 h. Bands were visualized by chemiluminiscence. Quantification of the bands was performed by densitometric analysis using Image J program (NIH) and was expressed as O.D. units in comparison to the expression of GAPDH that was used as loading control [24].

## Cell migration assay

To quantify the ability of tumor cells to migrate, an *in vitro* wound healing assay was performed according to previously described methods [25]. Cells ($1.5x10^5$/well) were seeded in 24-well plates with 0.5 ml of DMEM medium supplemented with 10% FBS and left to adhere. Then, cells were deprived of FBS to synchronize them. Then, the cell monolayer was scratched to produce a wound with a 200 μl pipette tip, washed twice with PBS and fresh culture medium without FBS containing different drugs was added. The migration of cells was photographed at regular intervals from the beginning of the assay during 28 h and the uncovered area was integrated with image J software (NIH). The results were expressed as the percentage of covered area. The conditions selected, allowed us to quantify cell migration, considering negligible cell division, since MDA-MB231 cell doubling time is longer than total experimental lasting for each assay.

## Tumor-induced angiogenesis

Female NUDE [N-NIH Nu/Nu(spf)] mice (3 months old and 25 g/mouse) were purchased from the animal facility of the Faculty of Veterinary Sciences, La Plata National University (La Plata, Argentine). These animals are frequently used in xenogeneic assays with human cells. Mice were kept under specific pathogen-free conditions following the protocol designed by the National Institute of Health (NIH, USA) in the Guidelines for the care and handling of laboratory animals (1986). They were kept in metallic cages (3 animals/cage) with a wood shaving bedding, changed every 48 h, in 12:12 h light:dark cycle at 21±2 ºC, with water and food *ad libitum*. Experimental procedures were approved by the Institutional Committee for the Care and Use of Laboratory Animals (CICUAL) from the School of Medicine, University of Buenos Aires (Protocol number 20020170100227BA).

Neovascularization induced by tumor cells, was analyzed using an *in vivo* bioassay previously described [26]. Briefly, MDA-MB231 cells were washed with PBS and detached with trypsin, and adjusted to a concentration of $3x10^6$ cells/ml in DMEM. Then, 0.1 ml of cell suspension were injected intradermically into the area surrounding the third mammary fat pad in each flank of mice under sterile conditions in the laboratory. Animals were treated intraperitoneally (i.p.) 24 h post-injection of cells, with carbachol (0.94 pg/mouse) or APE (4.66 μg/mouse) alone or in combination with PX (0.51 ng/mouse). Treatments were randomly assigned to each animal and were administered in three doses with 48 h intervals. Atropine (0.17 ng/mouse) or methoctramine (3.5 μg/mouse) were inoculated i.p. 20 min before other drugs. Drugs were dissolved in PBS under sterile conditions. Doses were calculated considering the concentrations added to cells *in vitro*, the amount of inoculated cells and the lasting of treatment. Each experiment includes 10 different treatments with 3 animals per treatment/group repeated 3 times.

On day 6, animals were sacrificed by $CO_2$ inhalation and the skin was exposed. Using a dissecting microscope (Konus Corporation, Miami, USA) at a 6.4 X magnification, the vascular response was examined in the inner surface of the skin. Pictures were taken of the inoculation sites with an incorporated digital camera (Canon Power Shot A75, Canon Inc., Lake Success, USA). Then, the pictures were projected onto a reticular screen to count the number of vessels per mm$^2$ of skin. Angiogenesis was quantified as vessel density, determined by the formula: Σ number of vessels in each square/total number of squares of each inoculation site [26].

## EC50 calculation

Using the GraphPad Prism 6 program, dose–response data were transformed, changed to percentage and fitted using to a sigmoidal curve following a maximal effective concentration (Emax) model with at least six data points. The EC50 and Emax values were obtained from this analysis. Only data with less than 20% in the coefficient of variation for EC50 values were considered.

## Statistical analysis

Results were expressed as mean ± S.E.M. The GraphPad Prism6 computer program was used employing one-way ANOVA analysis for paired samples to obtain the significance of differences between mean values in all control and test samples. The analysis was complemented by using a Tukey test to compare among mean values. Differences between means were considered significant if P<0.05. The data and statistical analysis complied with the recommendations on experimental design and analysis in pharmacology [27]. When it was necessary, values were also analyzed using Chou Talalay method to evaluate drug combination effects [28].

### Drugs

All drugs were purchased from Sigma Chemical Co. (St. Louis, USA) unless otherwise stated. Solutions were prepared fresh daily.

## Results

### Muscarinic agonists modify MDA-MB231 cell viability

In this work, we demonstrated by Western blot the expression of M$_1$, M$_2$, M$_4$ and M$_5$ receptors in MDA-MB231 cell lysates (Fig 1). The non-tumorigenic mammary cell line MCF-10A lacks all subtypes of muscarinic receptors (Fig 1).

In addition, we analyzed the effect of increasing concentrations of the non-selective muscarinic agonist carbachol added to MDA-MB231 cells in culture. Fig 2A shows that carbachol produced an inhibition in cell viability in a concentration-dependent manner (EC50: $1.2 \times 10^{-11}$M). Also APE was effective to reduce tumor cell viability at concentrations higher than $10^{-5}$M (EC50: $3.1 \times 10^{-5}$M) (Fig 2B). Due to the absence of muscarinic receptors in non-tumorigenic MCF-10A cells, neither carbachol nor APE modified cell viability at any concentration tested (Fig 2A and 2B). We confirmed the cytotoxic activity of PX on breast cancer cells. In our system, this chemotherapeutic agent produced a decrement in cell viability at concentrations $\geq 10^{-8}$M, when it was added to tumor cells (EC50: $8.5 \times 10^{-7}$M). As an undesirable action, PX also reduced MCF-10A cell viability from $10^{-7}$M (Fig 2C).

### Effects produced by the combined treatment of paclitaxel with muscarinic agonists on MDA-MB231 cells

In order to determine the ability of muscarinic agonists to synergize the action of PX on tumor cells, we performed concentration-response curves of this drug in the presence of the EC25 of carbachol or APE ($8.6 \times 10^{-12}$M or $1.1 \times 10^{-5}$M respectively) to evaluate cell viability (Fig 3). The addition of carbachol shifted to the left the dose-response curve of PX modifying the EC50

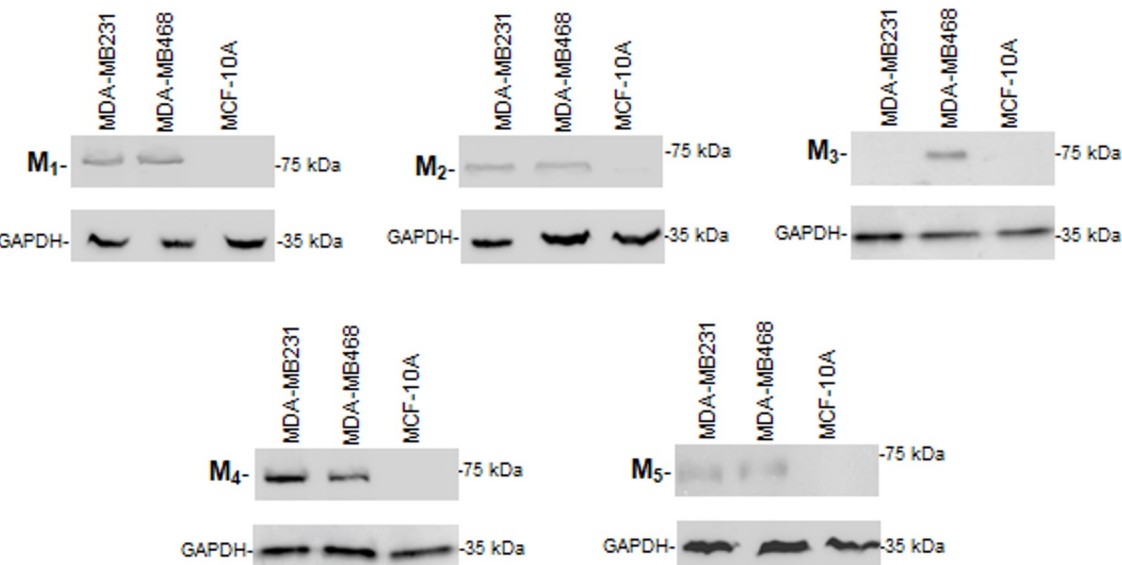

**Fig 1. Muscarinic receptors´ expression.** Western blot assay to detect muscarinic (M) receptor subtypes in MDA-MB231, MDA-MB468 or MCF-10A cells. Molecular weights are indicated on the right. The expression of glyceraldehyde 3-phosphate dehydrogenase (GAPDH) protein was used as loading control. One representative experiment of 3 is shown.

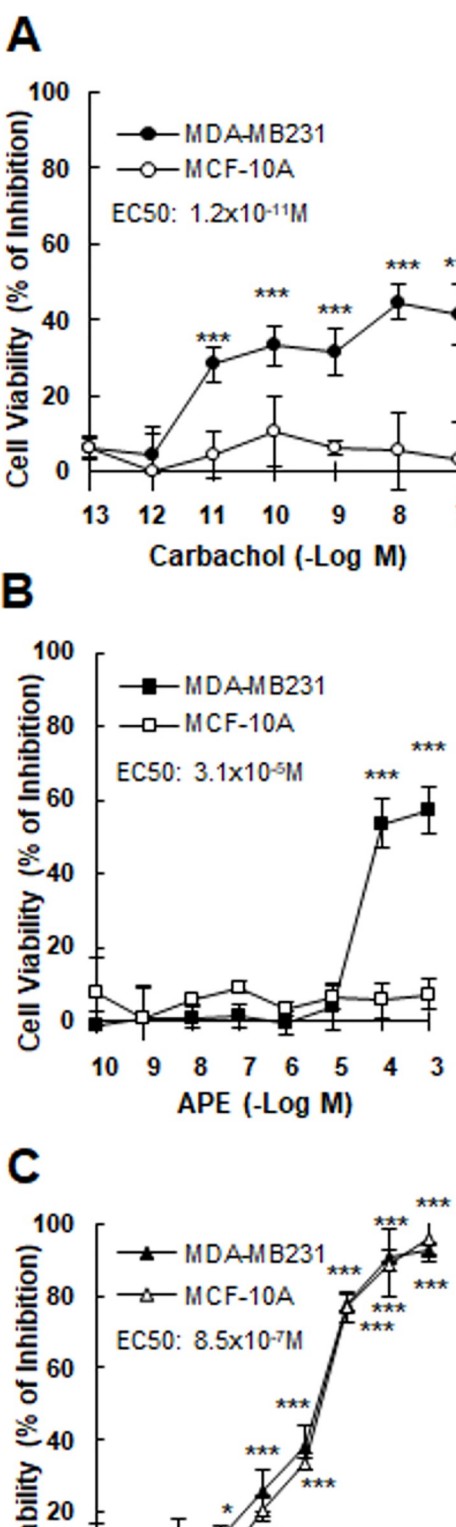

**Fig 2. Effect of muscarinic agonists or paclitaxel on breast cell viability.** Concentration-response curves of A) carbachol, B) arecaidine propargyl ester (APE) or C) paclitaxel on MDA-MB231 cells or MCF-10A cells. Results were expressed as percent of inhibition in cell viability respect to control (cells without treatment). EC50: effective concentration that produces half maximal response. Values are mean ± S.E.M. of 5 experiments performed in duplicate. (*P<0.05; **P<0.001;***P<0.0001 vs. control: untreated cells).

value by more than one order of magnitude (EC50 PX: $8.5 \times 10^{-7}$M; EC50PX+carbachol: $1.1 \times 10^{-8}$M) (Fig 3A). Similar results were obtained when APE was added to the concentration-response curve of PX (EC50: PX+APE: $3.5 \times 10^{-8}$M) on tumor cells (Fig 3B). These results and the combination index (CI)<1 calculated by the Chou Talalay method using CompuSyn Software (PX+carbachol: 0.10077; PX+APE: 0.13951) indicated a synergism of potentiation for both pairs of drugs.

Taking into account the undesirable effect induced by chemotherapy at usual doses ($\geq 10^{-6}$M), we analyzed the ability of PX at the first effective concentration ($10^{-8}$M) combined with the EC25 of carbachol or APE on the viability of tumor cells in order to mimic the dosage of metronomic therapy (Fig 4). The combination of PX plus carbachol significantly reduced cell viability of MDA-MB231 cells. This effect was prevented in the presence of $10^{-9}$M atropine, a non-selective muscarinic antagonist. This combination did not exert any action on non-tumorigenic MCF-10A mammary cells (Fig 4A). Considering that $M_2$ receptors are expressed in tumor cells, we analyzed the effect of the $M_2$ selective agonist APE in combination with PX. The addition of APE ($1.1 \times 10^{-5}$M) combined with PX potentiated the effect of PX alone to reduce tumor cell viability (Fig 4B). The action of APE plus PX was reverted by the previous addition of the $M_2$ selective antagonist methoctramine ($10^{-5}$M) revealing the main participation of this receptor subtype in the reduction of tumor cell viability (Fig 4B). In addition, we confirmed the participation of $M_2$ receptor by pretreating cells with a specific $M_2$-siRNA that prevented the action of APE plus PX on MDA-MB231 cell viability (S1 Fig). We also proved that the addition of carbachol, APE and/or PX at the same concentrations did not modify MCF-10A cell viability (Fig 4A and 4B).

To confirm that muscarinic agonists can synergize the action of another cytotoxic drug reducing tumor cell viability, we tested the effect of $10^{-8}$M doxorubicin (the first effective concentration) a drug frequently used in breast cancer treatment combined with carbachol or APE both at EC25 on MDA-MB231 cells (Table 1).

Either the presence of carbachol or APE potentiated the effect of doxorubicin by reducing tumor cell viability. The effect was prevented by preincubating cells with atropine or methoctramine. Moreover, in MDA-MB468 cells derived from another subtype of TN tumor, we observed that the addition of carbachol or APE at the EC25 ($1.1 \times 10^{-10}$M or $1.3 \times 10^{-7}$M respectively) to the first effective concentration of PX ($10^{-9}$M) was also effective to reduce cell viability. These tumor cells also express all subtypes of muscarinic receptors (Fig 1). The addition of atropine or methoctramine before the combined treatment significantly reduced this effect (Table 2).

We confirmed that the combination of PX at the first effective concentration plus carbachol or APE at the EC25 added in three cycles potently reduced cell viability of MDA-MB231 cells by more than 60%. This effect was prevented by the previous addition of atropine o methoctramine respectively (Fig 5A and 5B).

In order to investigate the mechanism of action involved in the effect produced by the combinations, we analyzed the expression of ABCG2 and EGFR in tumor cells. The addition of PX with carbachol (Fig 6A) or APE (Fig 6B) significantly reduced the expression of ABCG2 (PX plus carbachol P = 0.0270; PX plus APE P = 0.0425 vs. control). The addition of carbachol, APE or PX alone did not modify the expression of this transporter in tumor cells (Fig 6A and 6B).

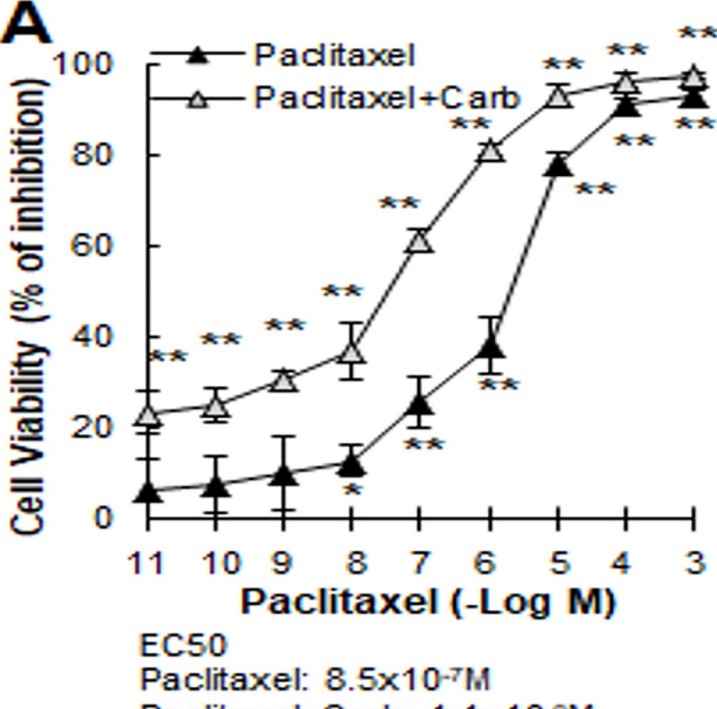

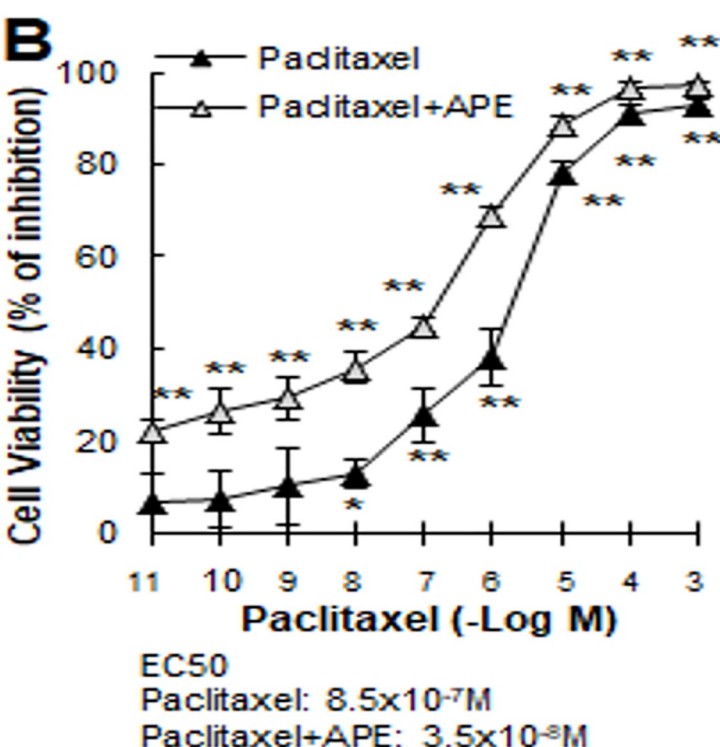

**Fig 3. Effect of muscarinic agonists on the concentration-response curve of paclitaxel.** The MDA-MB231 cells were treated with increasing concentrations of paclitaxel in the absence or presence of A) carbachol ($8.6 \times 10^{-12}$M) or B)

arecaidine propargyl ester (APE) ($1.1 \times 10^{-5}$M) during 48 h. Results were expressed as percent of inhibition in cell viability respect to control (cells without treatment). EC50: effective concentration that produces half maximal response. Values are mean ± S.E.M. of 5 experiments performed in duplicate. (*P<0.05; **P<0.0001 vs. control: untreated cells).

It has been documented that the constitutively activation of EGFR or its transactivation could contribute to drug resistance in different types of tumor cells [29]. As it is shown in Fig 6C, MDA-MB231 cells express this receptor and the addition of three cycles of PX plus carbachol significantly reduced protein expression by more than 40% in comparison to control (P = 0.0477) (Fig 6C). Similarly, Western blot analysis showed that the treatment with APE combined with PX also down-regulated EGFR protein by more than 41% respect to control (P = 0.0460) (Fig 6D).

### Effect produced by the combined treatment of muscarinic agonists and paclitaxel on MDA-MB231 cell migration

Since invasion is an important step in tumor progression, we analyzed the effect of PX plus carbachol or APE on tumor cell migration in an *in vitro* wound healing assay (Fig 7). At the end of experimental time (28 h) control wound was covered by 79±8% while the addition of PX plus carbachol or APE prevented wound covering by 42±4% or 42±6% respectively. Both effects were reverted in the presence of atropine or methoctramine.

### The combined treatment reduces vascular endothelial growth factor-A and tumor induced-angiogenesis

Taking into account that metronomic administration of drugs can exert additional benefits in comparison to traditional chemotherapy, we analyzed the ability of both drug combinations to exert anti-angiogenic actions. The addition of PX plus carbachol to MDA-MB231 cells in culture for three cycles significantly down-regulated VEGF-A expression by almost 39% (P = 0.0490) (Fig 8A). Moreover, the addition of APE to the combination exerted a similar reduction in VEGF-A expression in MDA-MB231 cells (37±3%) (P = 0.0257) (Fig 8B). We also analyzed the ability of MDA-MB231 cells to induce blood vessel formation in the skin of NUDE mice (Fig 8C). The inoculation of tumor cells increased by 31±5% (P = 0.0362) skin neovascularization in comparison to sham skin. The i.p. administration of carbachol plus PX at metronomic doses to tumor bearers for three cycles potently reduced tumor induced-angiogenesis. This effect was reverted when animals were treated i.p. with the non-selective antagonist atropine previously to the combination. Also, APE combined with PX at low doses significantly reduced tumor neovascularization and the pre-treatment of tumor bearers with methoctramine, a selective $M_2$ antagonist, prevented the latter effect. Neovascular response in the skin of tumor bearing mice treated with PX plus carbachol or APE did not differ from the vascular density exhibited by skin obtained from sham animals (3.07±0.10). Representative photographs of sham skin, positive neovascularized skin induced by MDA-MB231 cells (Control) and the inhibition of angiogenic response produced by the *in vivo* treatment of NUDE mice with PX plus carbachol or APE are shown in Fig 8D.

### Discussion and conclusions

In this work, we demonstrated for the first time, the expression of different muscarinic receptor subtypes (1, 2, 4 and 5) in MDA-MB231 cells derived from a human TN mammary adenocarcinoma. We also observed that the long-term addition of carbachol or APE (non-selective

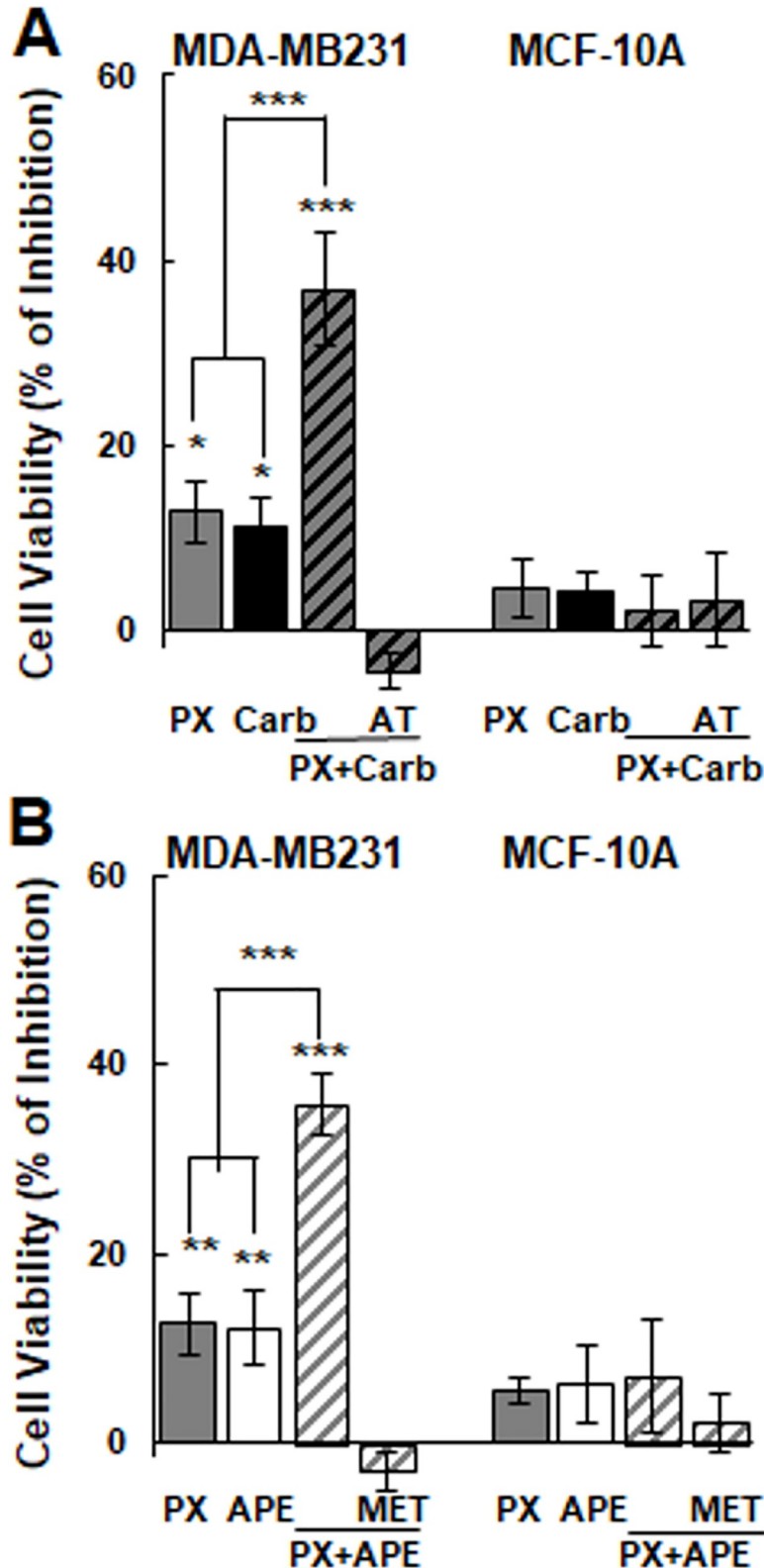

**Fig 4. Effect of the combination of paclitaxel with a muscarinic agonist on breast cell viability.** Cells were treated with A) paclitaxel (PX) ($10^{-8}$M) combined with carbachol (Carb) ($8.6 \times 10^{-12}$M) in the absence or presence of atropine (AT) ($10^{-9}$M) or B) PX ($10^{-8}$M) was combined with arecaidine propargyl ester (APE) ($1.1 \times 10^{-5}$M) in the absence or

presence of methoctramine (MET) ($10^{-5}$M). Results are expressed as percent of inhibition in cell viability respect to control (cells without treatment). Values are mean ± S.E.M. of 5 experiments performed in duplicate. (*P<0.01; **P<0.001; ***P<0.0001 vs. control, PX or Carb).

and $M_2$ receptor selective agonist, respectively) reduced cell viability in these tumor cells. Previous results from our group and also from other authors pointed to the ability of these muscarinic agonists to produce cell death in murine breast, bladder and neuronal tumor cells [17–19]. These results allowed us to consider muscarinic receptors as specific therapeutic targets for the treatment of breast tumors, in particular TN since they are classified as very aggressive and malignant. Previous reports indicated that TN tumor bearers benefit from the addition of PX in the adjuvant therapy, supporting the conclusion that taxanes are useful for this group of patients [5, 30]. Our results confirmed the ability of PX to decrease TN breast tumor cell viability, but it also produced similar actions on normal MCF-10A breast cells as an undesirable effect. The latter is one of the reasons why specific adjuvant regimens for TN breast cancer are still under revision, and third generation chemotherapy regimens utilizing dose dense or metronomic poly-chemotherapy are thought to be more effective and less harmful than traditional chemotherapy. Considering the latter, together with our previous findings in MCF-7 cells that were sensible to a low dose metronomic therapy focused on muscarinic receptors [16] we designed a metronomic administration of PX plus muscarinic agonists at low concentrations. This strategy, besides focusing on muscarinic receptors as novel targets for the treatment of TN tumors, could be less aggressive for normal mammary cells that lack of these receptors [2, 12]. Moreover, our results reveal the possibility of generalizing the usage of this combination of drugs since it is also effective when PX is replaced by doxorubicin. It is also important to consider that the administration of PX plus carbachol or APE is effective on MDA-MB468 cells, derived from a different TN tumor than MDA-MB231 classified as Basal A with amplified EGFR [31].

It is important to note that $M_2$ receptor mediates the main cytotoxic action of this low dose combined therapy, since similar results were obtained by replacing carbachol with APE, a selective $M_2$ agonist, in the combination. The participation of this receptor subtype was confirmed by preventing the effect of the combination either with the selective $M_2$ antagonist

**Table 1. Effect of the combination of doxorubicin with a muscarinic agonist on MDA-MB231 cell viability.**

| MDA-MB231 | |
|---|---|
| **Treatment** | **Cell Viability (% of Inhibition)** |
| DX + Carb | **35.3±0.8** ** |
| DX + Carb + AT | -6.7±2.1 |
| Carb | 11.2±3.2 * |
| DX | 12.4±4.6 * |
| DX + APE | **33.3±2.1** ** |
| DX + APE + MET | -7.7±2.6 |
| APE | 12.2±3.9 * |

Cells were treated with doxorubicin (DX) ($10^{-8}$M) combined with carbachol (Carb) ($8.6 \times 10^{-12}$ M) or with arecaidine propargyl ester (APE) ($1.1 \times 10^{-5}$ M) during 48 h in the absence or presence of atropine (AT) ($10^{-9}$M) or methoctramine (MET) ($10^{-5}$M) respectively. Results are expressed as cell viability (% of inhibition) respect to control (cells without treatment). Values are mean ± S.E.M. of 5 experiments performed in duplicate.

*P<0.001

**P<0.0001 vs. control (cells without treatment).

**Table 2. Effect of the combination of paclitaxel with a muscarinic agonist on MDA-MB468 cell viability.**

| MDA-MB468 | |
| --- | --- |
| Treatment | Cell Viability (% of Inhibition) |
| PX + Carb | **33.4±2.5** ** |
| PX + Carb + AT | -1.0±1.9 |
| Carb | 11.6±2.8 * |
| PX | 13.8±2.7 * |
| PX + APE | **26.9±3.6** ** |
| PX + APE + MET | 11.6±5.5 |
| APE | 9.1±2.2 * |

Cells were treated with paclitaxel (PX) ($10^{-9}$M) combined with carbachol (Carb) ($1.2 \times 10^{-10}$ M) or with arecaidine propargyl ester (APE) ($1.4 \times 10^{-7}$M) during 48 h in the absence or presence of atropine (AT) ($10^{-9}$M) or methoctramine (MET) ($10^{-5}$M) respectively. Results are expressed as cell viability (% of inhibition) respect to control (cells without treatment). Values are mean ± S.E.M. of 5 experiments performed in duplicate.

*P<0.001

**P<0.0001 vs. control (cells without treatment).

methoctramine or by transfecting cells with a specific M$_2$-siRNA. In line with our findings, similar effects were demonstrated in human glioblastoma cell lines and in glioblastoma cancer stem cells [19, 20].

One of the most important side effects of traditional chemotherapy is the resistance to drugs. In fact, the most commonly observed mechanism conferring drug resistance in cancer cells is the over-expression of ABC transporters that mediates the efflux of endogenous and exogenous substances using energy provided by ATP hydrolysis. ABCG subfamily plays pivotal roles in the transport of anticancer drugs out of cells, generating the development of drug resistance [32]. Here we analyzed the expression of ABCG2 since this protein is expressed in CD44$^+$/CD24$^-$ stem cell population that is abundant in MDA-MB231 tumor. The ABCG2 expression was potently down-regulated by the combination of PX either with carbachol or with APE. In relation with our results, Farhana et al. reported that malignancy of colon cancer cells induced by bile acids promotes cancer stemness in colonic epithelial cells by up-regulating M$_3$ receptor together with ABCG2 expression [33].

Several aggressive tumors present an overexpression of the HER2 marker considering it a target in cancer therapy [34, 35]. A variant with high homology to this marker is the EGFR which is also over-expressed in cancer [36, 37]. Its expression and/or activity can be modulated by muscarinic receptors [38]. Several authors consider that the sole up-regulation of EGFR expression is a bad prognosis marker and an indicator of chemoresistance in TNBC patients or cell lines [39–41]. Our results demonstrated that the combination of PX plus carbachol or APE drastically reduced EGFR expression in tumor cells and this effect should be beneficial in the treatment of this type of tumor. Recently, it has been described that the inhibition of EGFR activity in glioblastomas produces a decrement in drug resistance activity mediated by the ATP-dependent membrane transporter, ABC [42]. Although we only evaluated a down-regulation in the expression of EGFR due to drug treatment, more experiments could be useful to evaluate its activity in our system.

It should be important to consider that, the reduction in cell viability produced by the combination of PX with carbachol or APE could be related to a decrement in the number of cancer stem cells which are present at very high levels in TN tumors and are generally positive to ABCG2 transporter as it was previously reported [43, 44]. In addition, the over-expression/

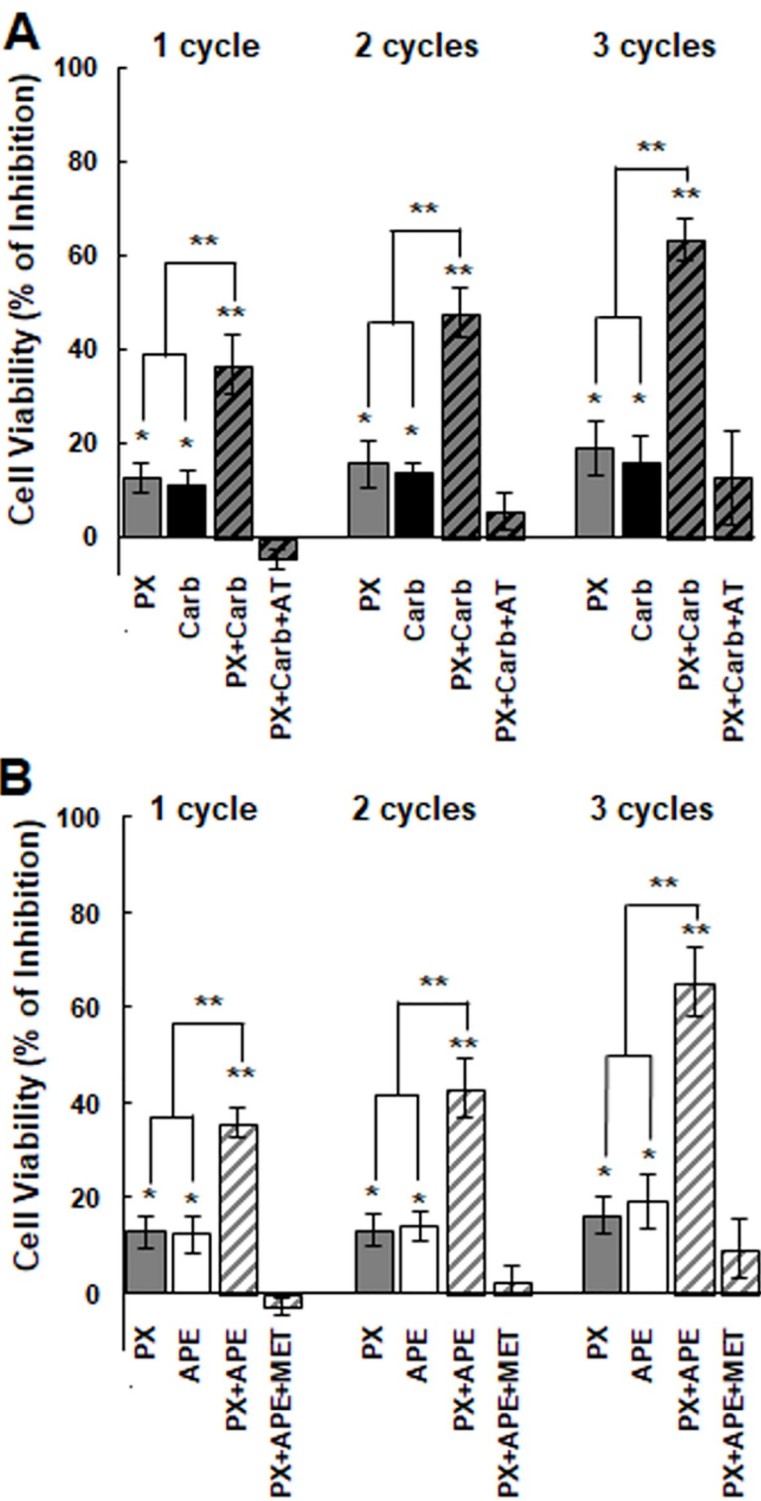

**Fig 5. Effect of the combination of paclitaxel with a muscarinic agonist on breast cell viability administered in three cycles.** Cells were treated with A) paclitaxel (PX) ($10^{-8}$M) combined with carbachol (Carb) ($8.6 \times 10^{-12}$M) in the absence or presence of atropine (AT) ($10^{-9}$M) or B) PX ($10^{-8}$M) was combined with arecaidine propargyl ester (APE) ($1.1 \times 10^{-5}$M) in the absence or presence of methoctramine (MET) ($10^{-5}$M). Results are expressed as percent of inhibition in cell viability respect to control (cells without treatment). Values are mean ± S.E.M. of 4 experiments performed in duplicate. (*P<0.05; **P<0.01***P<0.001; ****P<0.0001 vs. control, PX or Carb).

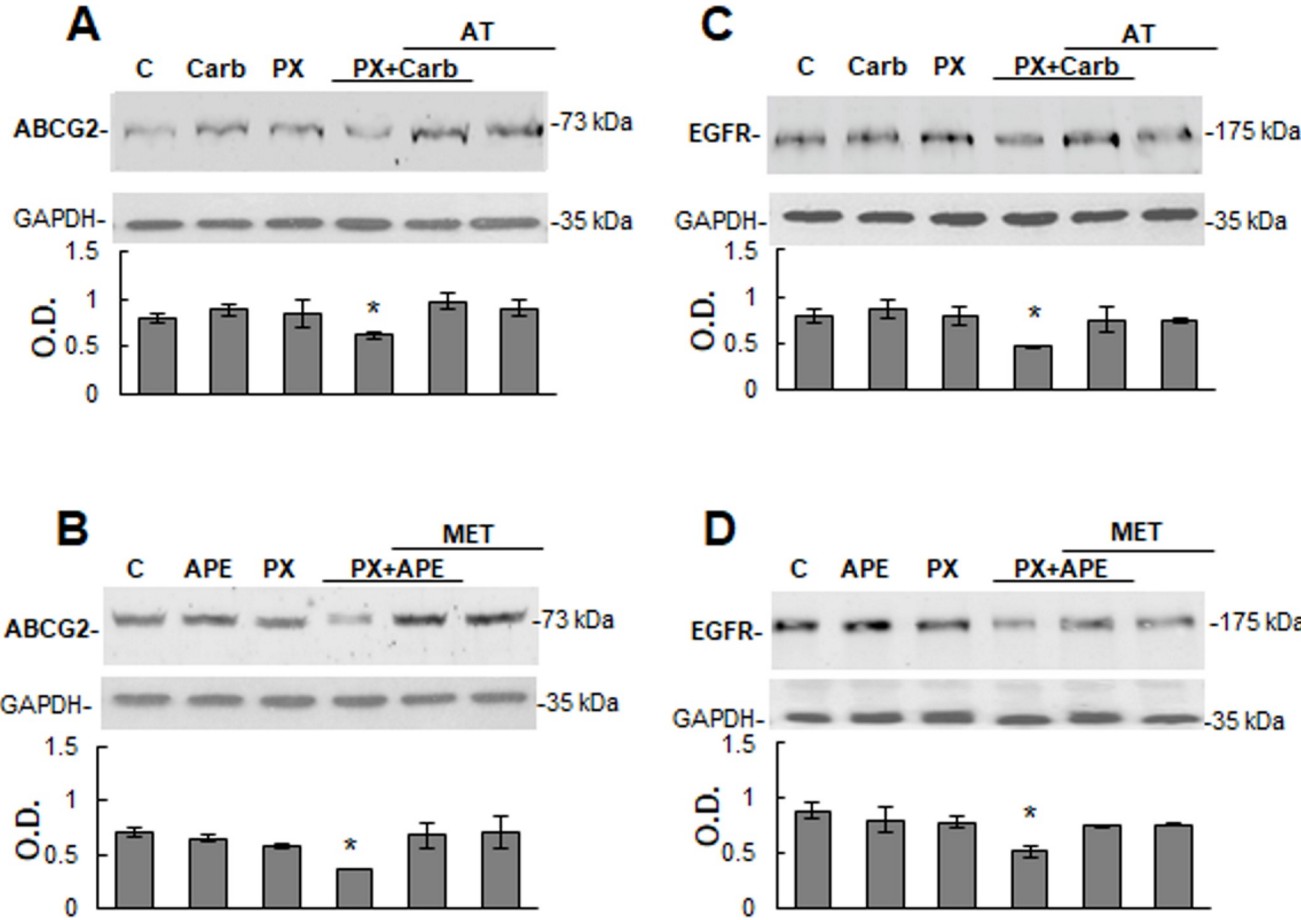

**Fig 6. Expression of ATP binding cassette G2 transporter and epidermal growth factor receptor in MDA-MB231 cells.** The expression of ATP binding cassette G2 (ABCG2) and epidermal growth factor receptor (EGFR) in tumor cells was analyzed by Western blot. Cells were treated for three cycles with paclitaxel (PX) ($10^{-8}$M) combined with A) and C) carbachol (Carb) ($8.6 \times 10^{-12}$M) or with B) and D) arecaidine propargyl ester (APE) ($1.1 \times 10^{-5}$M) in the absence or presence of atropine (AT) ($10^{-9}$M) or methoctramine (MET) ($10^{-5}$M). Molecular weights are indicated on the right. Densitometric analysis of the bands was expressed as optical density (O.D.) units relative to the expression of glyceraldehyde 3-phosphate dehydrogenase (GAPDH) protein used as loading control. One representative experiment of 3 is shown ($^*$P<0.05 vs. control).

activation of EGFR leads to VEGF-A production and is closely related to the proliferative behavior of breast cancer cells as well as tumor endothelial cells [45].

Invasion and/or migration are the most important steps in tumor progression since they are linked to metastasis and aggressiveness that are usually observed in TN breast cancer patients. Our results show for the first time the ability of low dose combined therapy to reduce tumor cell migration targeting M$_2$ receptor. In addition, angiogenesis has been broadly considered as a switch-on mechanism in tumor growth and metastasis. One of the additional benefits of metronomic therapy to be tested is its ability to reduce tumor-induced angiogenesis [46]. Here we confirmed that the treatment of TN tumor cells with the combination of PX plus muscarinic agonists is able to reduce the expression of VEGF-A, and also the *in vivo* neovascular response induced by tumor cells. The treatment of tumor cells with the combination of PX plus muscarinic agonists produced a decrement in VEGF-A levels *in vitro*. We cannot discard that the anti-angiogenic actions observed *in vivo* could also be due to a decrement in the number of tumor cells, or to a down-regulation of other angiogenic factors produced by tumor cells and/or by stromal cells. More experiments are needed to clarify these aspects in our model.

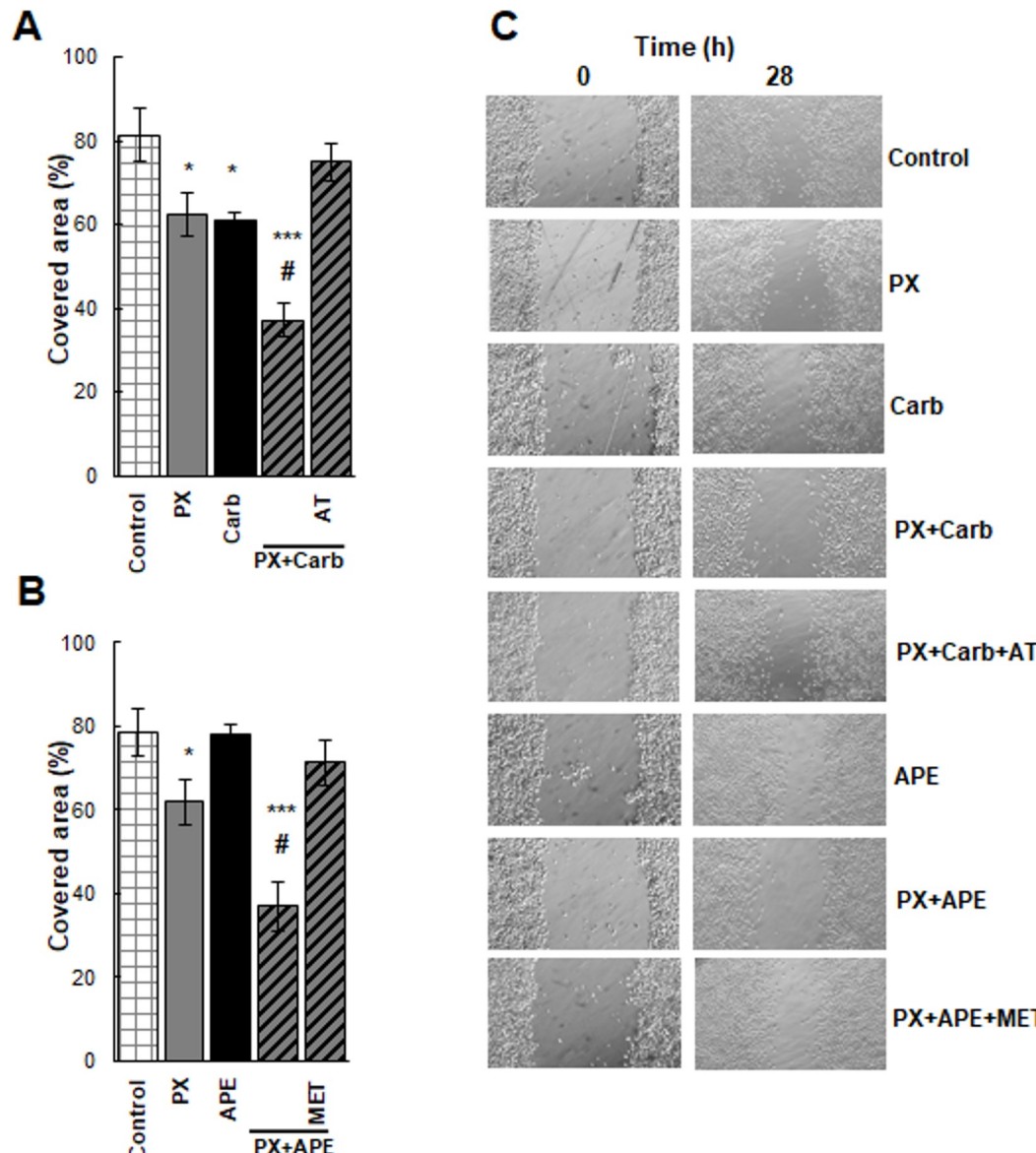

**Fig 7. Effect of the combination of paclitaxel with a muscarinic agonist on MDA-MB231 cell migration.** Cell migration was evaluated as the percentage of covered area after 28 h of treatment with A) paclitaxel (PX) ($10^{-8}$M) combined with carbachol (Carb) ($8.6 \times 10^{-12}$M) in the absence or presence of atropine (AT) ($10^{-9}$M) or B) PX ($10^{-8}$M) combined with arecaidine propargyl ester (APE) ($1.1 \times 10^{-5}$M) in the absence or presence of methoctramine (MET) ($10^{-5}$M). C) Representative photographs (64X) were obtained by phase contrast at the beginning (0h) and at the end of experimental time (28 h). Values are mean ± S.E.M. of 3 experiments performed in duplicate. ($^{*}$P<0.05; $^{**}$P<0.01; $^{***}$P<0.0001 vs. control; #P<0.001 vs. each drug added alone).

In conclusion, our results demonstrate that low doses therapy combining PX with a non-selective or a selective $M_2$ agonist could be a useful strategy to treat TN breast tumors. In particular, the treatment focused on $M_2$ receptor appears as a new promising therapeutic target to counteract not only breast cancer cell survival, but also to reduce invasion and pathological neo-angiogenesis, suggesting a possible prevention of chemoresistance by reducing ABCG2 and EGFR expression.

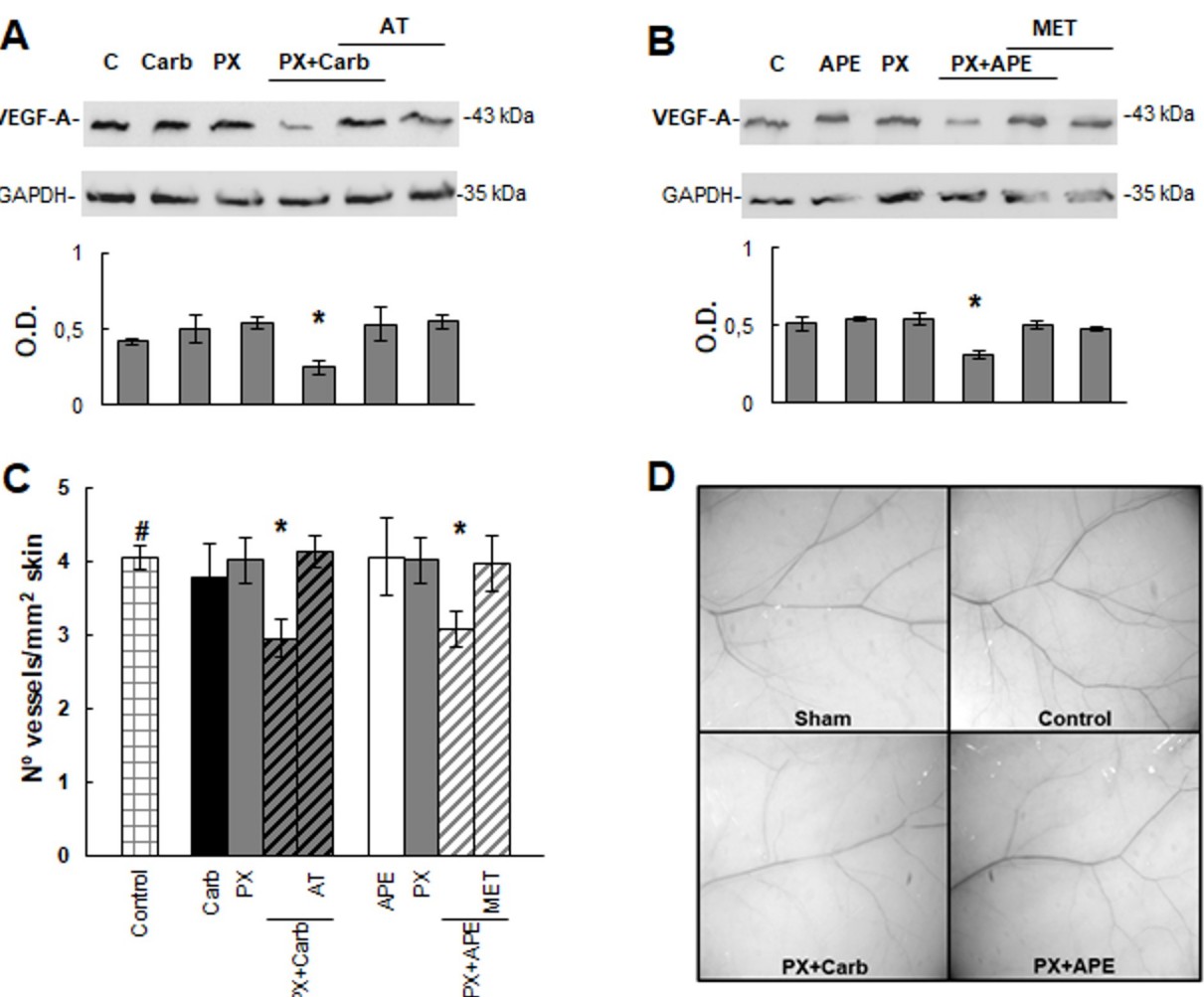

**Fig 8. Tumor induced angiogenesis.** To analyze the expression of vascular endothelial growth factor-A (VEGF-A) by Western blot, MDA-MB231 cells were treated with A) paclitaxel (PX) ($10^{-8}$M) combined with carbachol (Carb) ($8.6 \times 10^{-12}$M) or B) with arecaidine propargyl ester (APE) ($1.1 \times 10^{-5}$M) in the absence or presence of atropine (AT) ($10^{-9}$M) or methoctramine (MET) ($10^{-5}$M) respectively. Molecular weights are indicated on the right. Densitometric analysis of the bands was expressed as optical density (O.D.) units relative to the expression of glyceraldehyde 3-phosphate dehydrogenase (GAPDH) protein used as loading control. One representative experiment of 3 is shown. C) *In vivo* neovascularization induced by MDA-MB231 cells in NUDE mice skin. Cells were inoculated as it was stated in Methods, and drugs were administered i.p. Values are mean ± S.E.M. of 3 experiments performed with 3 animals per group inoculated in both flanks. D) Representative photographs of mice skin from sham animals or inoculated with tumor cells (Control) without treatment or treated with PX+Carb or PX+APE. Magnification 6.4X. (#P<0.05 vs. sham skin; *P<0.05 vs. control).

## Supporting information

**S1 Checklist. The ARRIVE guidelines checklist.**
(DOCX)

**S1 Fig. Muscarinic subtype 2 receptor expression silencing.** A) Western blot assay to detect muscarinic (M) receptor subtype 2 in MDA-MB231 cells in the absence or presence of NC-siRNA or $M_2$-siRNA. Molecular weights are indicated on the right. Densitometric analysis of the bands is expressed as optical density (O.D.) units relative to the expression of glyceraldehyde 3-phosphate dehydrogenase (GAPDH) protein used as loading control. One representative experiment of 3 is shown. (*P<0.01 vs. cells without $M_2$-siRNA). B) Effect of the combination of paclitaxel (PX) ($10^{-8}$M) with arecaidine propargyl ester (APE) ($1.1 \times 10^{-5}$M) in

the absence or presence of M$_2$-siRNA. Results are expressed as percent of inhibition in cell viability respect to control (cells without treatment). Values are mean ± S.E.M. of 3 experiments performed in duplicate. ($^*$P<0.05 $^{**}$P<0.001 vs. control or PX).
(TIF)

**S1 Raw images.**
(PDF)

**S2 Raw images.**
(PDF)

**S1 Dataset.**
(XLS)

**S2 Dataset.**
(XLSX)

## Acknowledgments

The authors want to thank Mr. Francisco Sanchez, Mr. Daniel Gonzalez and Vet. Marcela Vázquez for their excellent technical assistance and to Mrs. Patricia Fernández for her excellent management of financial support. We also want to thank Dr. Federico Penas for providing NC-siRNA, Dr. Fernanda Troncoso and Lic. Vanina Vacheta from INQUIFIB-CONICET for providing MDA-MB468 cell line.

## Author Contributions

**Conceptualization:** Ada M. Tata, María E. Sales.

**Data curation:** Alejandro J. Español, Agustina Salem.

**Formal analysis:** Alejandro J. Español, Agustina Salem.

**Funding acquisition:** Ada M. Tata, María E. Sales.

**Investigation:** Alejandro J. Español, Agustina Salem, María Di Bari, Ilaria Cristofaro, Yamila Sanchez.

**Methodology:** María Di Bari, Ilaria Cristofaro, Yamila Sanchez.

**Project administration:** Ada M. Tata, María E. Sales.

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
