## [Decision Letter · Decision Letter 0]

4 Feb 2020

PONE-D-19-32678

The metronomic combination of paclitaxel with cholinergic agonists inhibits triple negative breast tumor progression. Participation of M2 receptor subtype

PLOS ONE

Dear Dr. Espanol,,

Thank you for submitting your manuscript to PLOS ONE. After careful consideration, we feel that it has merit but does not fully meet PLOS ONE’s publication criteria as it currently stands. Therefore, we invite you to submit a revised version of the manuscript that addresses the points raised during the review process, particularly on presentation of the data in Figures.

We would appreciate receiving your revised manuscript by Mar 20 2020 11:59PM. To enhance the reproducibility of your results, we recommend that if applicable you deposit your laboratory protocols in protocols.io, where a protocol can be assigned its own identifier (DOI) such that it can be cited independently in the future. For instructions see: http://journals.plos.org/plosone/s/submission-guidelines#loc-laboratory-protocols

We look forward to receiving your revised manuscript.

Kind regards,

Surinder K. Batra

Academic Editor

PLOS ONE

Journal Requirements:

1. We noticed you have some minor occurrence(s) of overlapping text with the following previous publication(s), which needs to be addressed:

https://doi.org/10.2174/1381612822666160229115317

https://doi.org/10.1016/j.ctrv.2016.09.004

https://doi.org/10.1016/j.ejphar.2012.03.013

https://doi.org/10.1016/j.intimp.2015.03.018

https://doi.org/10.1111/bph.12834

https://doi.org/10.1371/journal.pone.0057572

https://doi.org/10.1080/01635581.2017.1328605

http://dx.doi.org/10.3748/wjg.v22.i30.6876

In your revision ensure you cite all your sources (including your own works), and quote or rephrase any duplicated text outside the Methods section. Further consideration is dependent on these concerns being addressed.

2. As part of your revision, please complete and submit a copy of the ARRIVE Guidelines checklist, a document that aims to improve experimental reporting and reproducibility of animal studies for purposes of post-publication data analysis and reproducibility: https://www.nc3rs.org.uk/arrive-guidelines. Please include your completed checklist as a Supporting Information file. Note that if your paper is accepted for publication, this checklist will be published as part of your article.

3.

In your Data Availability statement, you have not specified where the minimal data set underlying the results described in your manuscript can be found. PLOS defines a study's minimal data set as the underlying data used to reach the conclusions drawn in the manuscript and any additional data required to replicate the reported study findings in their entirety. All PLOS journals require that the minimal data set be made fully available. For more information about our data policy, please see http://journals.plos.org/plosone/s/data-availability.

4.

PLOS ONE now requires that authors provide the original uncropped and unadjusted images underlying all blot or gel results reported in a submission’s figures or Supporting Information files. This policy and the journal’s other requirements for blot/gel reporting and figure preparation are described in detail at https://journals.plos.org/plosone/s/figures#loc-blot-and-gel-reporting-requirements and https://journals.plos.org/plosone/s/figures#loc-preparing-figures-from-image-files. When you submit your revised manuscript, please ensure that your figures adhere fully to these guidelines and provide the original underlying images for all blot or gel data reported in your submission. See the following link for instructions on providing the original image data: https://journals.plos.org/plosone/s/figures#loc-original-images-for-blots-and-gels.

Reviewers' comments:

Reviewer's Responses to Questions

**Comments to the Author**

1. Is the manuscript technically sound, and do the data support the conclusions?

Reviewer #1: Yes

Reviewer #2: Yes

2. Has the statistical analysis been performed appropriately and rigorously? 

Reviewer #1: Yes

Reviewer #2: Yes

3. Have the authors made all data underlying the findings in their manuscript fully available?

Reviewer #1: Yes

Reviewer #2: Yes

4. Is the manuscript presented in an intelligible fashion and written in standard English?

Reviewer #1: Yes

Reviewer #2: No

5. Review Comments to the Author

Reviewer #1: In this study authors demonstrate the role of metronomic therapy employing a traditional anti-cancer drug, paclitaxel plus muscarinic agonists at low doses on Triple negative tumors. In general, this study is well-performed and qualified to accept to PLOS ONE for publication if the authors address two major comments as below:

1. It is recommended to run all the samples of proteins in the single gel in Figure-1A and Figure-1B and present the westernblot data together and not individually.

2. GAPDH levels for MCF10A cells are not comparable with MDAMB231 cells. It seems that the amount of total protein is less in the MCF10 cells thus the comparison M-receptors are not accurate. This experiment needs to be done more convincingly.

Reviewer #2: The manuscript entitled “The metronomic combination of paclitaxel with cholinergic agonists inhibits triple-negative breast tumor progression. Participation of M2 receptor subtype”. Espanol AJ and colleagues were trying to identify the role of muscarinic receptors on cancer cell growth and migration. In this study, authors used carbachol or arecaidine propargyl ester along with traditional anti-cancer drugs paclitaxel and their inhibitory effect on breast cancer cells. Also, they have shown the chemoresistance mechanism due to the overexpression of ABCG2 and EGFR expression. EGFR activation is more important than just expression, moreover, EGFR signaling is highly activated other breast cancer subtypes rather than TNBC. Authors need to provide full blot rather than crop in Figure 1 and sup Fig.1. Bar diagram would be a better representation of the quantification of wound healing experiments (Figure 7A and B). In Figure 8A and B, the expression of VEGF is not showing any clear difference.

6. PLOS authors have the option to publish the peer review history of their article (what does this mean?). If published, this will include your full peer review and any attached files.

Reviewer #1: No

Reviewer #2: No

---

## [Author Response · Author response to Decision Letter 0]

13 Mar 2020

We thank the editor and reviewers for your useful suggestions that helped our manuscript improve. 

RESPONSE TO ACADEMIC EDITOR AND REVIEWERS

1. We noticed you have some minor occurrences of overlapping text…

We rephrased or cited all duplicated text found in the manuscript.

We rephrased the following sections (line number corresponds to “Manuscript with track Changes”):

In Material and Methods; Cell culture: 

• We replaced “obtained” by “acquired”. Line 105.

• We replaced “purchased” by “bought”. Line 109.

• We replaced “detached” by “removed”. Line 113.

• We replaced “replaced” by “changed”. Line 115.

• We replaced “assayed” by “determined”. Line 116.

• We replaced “confirmed” by “observed”. Line 117.

In Material and Methods; Detection of muscarinic receptors by Western blot:

• We replaced paragraph from line 120 to line 135 by paragraph from line 136 to line 151.

In Material and Methods; Cell viability assay:

• We replaced paragraph from line 155 to line 175 by paragraph from line 176 to line 196.

In Material and Methods; Tumor-induced angiogenesis:

• We replaced paragraph from line 236 to line 255 by paragraph from line 256 to line 276.

The following references were added: 

• Sales ME, Español A, Salem A, Pulido P, Sanchez Y, Sanchez F. Role of muscarinic acetylcholine receptors in breast cancer. Design of metronomic chemotherapy. Curr Clin Pharmacol. 2019; 14: 91-100.

• Schettini F, Buono G, Cardelasi C, Desideri I, de Palacios S, Del Mastro L. Hormone receptor/human epidermal growth factor receptor 2-positive breast cancer: where we are now and where we are going. Cancer Treat Rev. 2016; 46: 20-26.

• Sales ME. Muscarinic receptors as targets for metronomic therapy in breast cancer. Curr Pharm Des. 2016; 22: 2170-2177.

• Dai X, Zhang J, Guo G, Cai Y, Cui R, Yin C, et al. A mono-carbonyl analog of curcumin induces apoptosis in drug-resistant and mitochondrial dysfunction. Cancer Manag Res. 2018; 10: 3069-3082.

• Hu T, Li Z, Gao CY, Cho CH. Mechanisms of drug resistance in colon cancer and its therapeutic strategies. World J Gastroenterol. 2016; 22: 6876-6889.

The following reference was removed: 

• Hayes DF, Thor AD, Dressler LG, Weaver D, Edgerton S, Cowan D, et al. Cancer and Leukemia Group B (CALGB) Investigators. HER2 and response to paclitaxel in node-positive breast cancer. N Engl J Med. 2007; 357:1496-506.

2. As part of your revision, please complete and submit a copy of the ARRIVE Guidelines checklist.

We attached the corresponding file named “ARRIVE Guidelines PONE-D-19-32678”.

3. Upon re-submitting your revised manuscript, please upload your study´s minimal underlying data set.

We attached the corresponding excel file named “S2 minimal data set”.

4. PLOS ONE now requires that authors provide the original uncropped and unadjusted images underlying all blot or gel results reported.

We attached the corresponding PDF file named “S1 raw images”.

5. Review Comments to the Author

Reviewer #1: In this study authors demonstrate the role of metronomic therapy employing a traditional anti-cancer drug, paclitaxel plus muscarinic agonists at low doses on Triple negative tumors. In general, this study is well-performed and qualified to accept to PLOS ONE for publication if the authors address two major comments as below:

1. It is recommended to run all the samples of proteins in the single gel in Figure-1A and Figure-1B and present the western blot data together and not individually.

Western blot assays for muscarinic receptors/GAPDH were done again and we made the Figure according reviewer´s suggestions (Fig 1).

2. GAPDH levels for MCF10A cells are not comparable with MDAMB231 cells. It seems that the amount of total protein is less in the MCF10 cells thus the comparison M-receptors are not accurate. This experiment needs to be done more convincingly.

See response in item 1, please.

Reviewer #2: The manuscript entitled “The metronomic combination of paclitaxel with cholinergic agonists inhibits triple-negative breast tumor progression. Participation of M2 receptor subtype”. Espanol AJ and colleagues were trying to identify the role of muscarinic receptors on cancer cell growth and migration. In this study, authors used carbachol or arecaidine propargyl ester along with traditional anti-cancer drugs paclitaxel and their inhibitory effect on breast cancer cells. Also, they have shown the chemoresistance mechanism due to the overexpression of ABCG2 and EGFR expression. 

1. EGFR activation is more important than just expression, moreover, EGFR signaling is highly activated in other breast cancer subtypes rather than TNBC. 

Even though that the activation of EGFR is important in the regulation of tumor growth; several authors consider that the up-regulation of EGFR expression is a bad prognosis marker and an indicator of chemoresistance in triple negative breast cancer (TNBC) patients or cell lines:

a. Zhang et al., Biomed Res Int. 2015;2015:357485 pointed to the distinct expression of EGFR and the prevalence of BRCA1 mutation indicating that EGFR, and BRCA1 might be unique biomarkers for targeted therapy and prognosis in TNBC.

b. Turner et al., Sci Rep. 2020 Jan 30;10(1):1493 concluded that EGFR is highly expressed in basal-like PDXs, cell lines, and patients, and high expression of this gene reduces metastasis-free survival, suggesting that targeting of this protein holds promise for potential clinical success in TNBC.

c. Islam et al., in J Cell Physiol. 2020 Jan 20. doi: 10.1002/jcp.29466 indicate that, the chemotolerance mechanism of TNBC has not yet been studied in detail. Infrequent messenger RNA expression, gene amplification (10-32.5%), and mutation (1%) of EGFR were seen in the TNBC samples irrespective of therapy, suggesting the importance of EGFR protein stabilization in this tumor. Thus, this study showed that reduced nuclear expression of Y654-p-β-catenin in NACT samples due to down-regulation of EGFR protein through promoter hypomethylation-mediated up-regulation of SH3GL2, resulting in low proliferation index/CD44 prevalence with better prognosis of the NACT patients, might have an important role in the chemotolerance of TNBC.

d. Fleisher et al., Breast Cancer (Dove Med Press). 2019 Jul 23; 11:231-241 postulated that EGFR is frequently overexpressed in TNBC, and the EGFR-overexpressing TNBC presumably escapes EGFR inhibitor therapy by up-regulating autophagy and inhibiting apoptosis. To parse the autophagy-apoptosis crosstalk pathway as a potential targeted therapy in TNBC, the activity of an EGFR inhibitor, osimertinib, alone and in combination with an autophagy inhibitor, chloroquine, was examined in EGFR-overexpressing TNBC cell line, MDA-MB-231.

2. Authors need to provide full blot rather than crop in Figure 1 and sup Fig.1. 

It is not possible to make Western blot assays using antibodies against the 5 muscarinic receptor subtypes in the same individual membrane, at the same time, because they have very similar molecular weights which makes difficult their identification. 

See response in item 1 to Reviewer 1, please.

3. Bar diagram would be a better representation of the quantification of wound healing experiments (Figure 7A and B).

We changed Figure 7A-B by bar graphics according reviewer´s suggestions.

4. In Figure 8A and B, the expression of VEGF is not showing any clear difference.

We made new Western blot assays according reviewer´s suggestions. The Images included in Fig 8A-B shows the differences among experimental groups.

---

## [Decision Letter · Decision Letter 1]

1 Jun 2020

PONE-D-19-32678R1

The metronomic combination of paclitaxel with cholinergic agonists inhibits triple negative breast tumor progression. Participation of M2 receptor subtype

PLOS ONE

Dear Dr. Español,

Thank you for submitting your manuscript to PLOS ONE. After careful consideration, we feel that it has merit but does not fully meet PLOS ONE’s publication criteria as it currently stands. Therefore, we invite you to submit a revised version of the manuscript that addresses the points raised during the review process.

It is necessary to determine not just the expression, but the activity of EGFR upon muscarinic receptor activation.

Additional titrating experiments are needed for APE and methoctramine.

Scratch assay is not an invasion assay but a migration assay. Please add the details of how this assay was performed (controls etc.)

Tumor angiogenesis assays were set up inappropriately. Angiogenesis requires the tumor to be formed, which process may take a few weeks in this particular strain.  Thus any changes in the vessel density observed within 6 days post injection may reflect the death of the injected cells.  Please modify you conclusions appropriately.

Finally, the manuscript has numerous typos and grammatical errors and would benefit significantly from scientific writer editing from a native English speaker.

We look forward to receiving your revised manuscript.

Kind regards,

Irina V. Lebedeva, Ph.D.

Academic Editor

PLOS ONE

Reviewers' comments:

Reviewer's Responses to Questions

**Comments to the Author**

1. If the authors have adequately addressed your comments raised in a previous round of review and you feel that this manuscript is now acceptable for publication, you may indicate that here to bypass the “Comments to the Author” section, enter your conflict of interest statement in the “Confidential to Editor” section, and submit your "Accept" recommendation.

Reviewer #1: All comments have been addressed

Reviewer #2: All comments have been addressed

Reviewer #3: (No Response)

Reviewer #4: All comments have been addressed

2. Is the manuscript technically sound, and do the data support the conclusions?

Reviewer #1: Yes

Reviewer #2: Yes

Reviewer #3: Partly

Reviewer #4: Partly

3. Has the statistical analysis been performed appropriately and rigorously? 

Reviewer #1: Yes

Reviewer #2: Yes

Reviewer #3: Yes

Reviewer #4: Yes

4. Have the authors made all data underlying the findings in their manuscript fully available?

Reviewer #1: Yes

Reviewer #2: Yes

Reviewer #3: Yes

Reviewer #4: Yes

5. Is the manuscript presented in an intelligible fashion and written in standard English?

Reviewer #1: Yes

Reviewer #2: Yes

Reviewer #3: Yes

Reviewer #4: Yes

6. Review Comments to the Author

Reviewer #1: (No Response)

Reviewer #2: All questions have been addressed in the revised manuscript entitled 'The metronomic combination of paclitaxel with cholinergic agonists inhibits triple negative breast tumor progression. Participation of M2 receptor subtype'

Reviewer #3: The authors have addressed most of comments from previous reviewers except one point from reviewer 2 about EGFR activation (see below). I have several other concerns that need to be addressed.

1. As with other G protein coupled receptors, muscarinic receptors can trans-activate EGFR, leading to increased cell proliferation and migration. Therefore, as pointed out by the reviewer 2, it is important to determine the activity of EGFR but not just its expression if the author attempts to link the downregulation of EGFR expression by muscarinic receptor activation to the inhibition of cancer cell growth and migration.

2. APE and methoctramine are not absolute muscarinic receptor M2-selective agonist and antagonist. With the concentrations used in the studies, they cannot distinguish which muscarinic receptor subtypes are involved in the effect of muscarinic receptor activation. The author has to either revise the manuscript to remove the description of specific involvement of the M2 receptor or perform additional experiments by titrating the dose of drugs used.

3. How was the wound healing assay performed? Was an inhibitor of cell proliferation included? If not, then the effect of treatment on wound healing may be secondary to its effect on cell proliferation.

4. The studies appear to be a replicate of the similar studies performed by the same group as noted in ref 16 and 17, except for using different cell lines. What are the new information provided in this manuscript?

Reviewer #4: The authors have addressed the reviews they received in the initial round of peer review to a satisfactory level.

7. PLOS authors have the option to publish the peer review history of their article (what does this mean?). If published, this will include your full peer review and any attached files.

Reviewer #1: No

Reviewer #2: No

Reviewer #3: No

Reviewer #4: No

---

## [Author Response · Author response to Decision Letter 1]

28 Jul 2020

We thank to the editor and reviewers for your useful suggestions that helped to improve our manuscript 

RESPONSE TO ACADEMIC EDITOR AND REVIEWERS

1.- It is necessary to determine not just the expression, but the activity of EGFR upon muscarinic receptor activation. 

As with other G protein coupled receptors, muscarinic receptors can trans-activate EGFR, leading to increased cell proliferation and migration. Therefore, as pointed out by the reviewer 2, it is important to determine the activity of EGFR but not just its expression if the author attempts to link the downregulation of EGFR expression by muscarinic receptor activation to the inhibition of cancer cell growth and migration.

Regarding transactivation of EGFR by G protein coupled receptors like that produced by muscarinic receptors, it is important to consider that:

The mechanism of EGFR transactivation triggered by cholinergic agonists like acetylcholine or carbachol has been studied mainly in colon cancer cells besides other normal cell lines. 

Most authors indicated that acetylcholine or carbachol exerts transactivation when added at concentrations ≥ 1uM and during short periods of time (0-60 min) to cultures. The last conditions are totally different from ours. Moreover, we did not observe transactivation when agonists, carbachol or APE was added alone, since they did not modify tumor cell viability.

In addition, transactivation was frequently due to an activation or up-regulation of M3 receptor subtype, which is absent in MDA-MB231 cells. 

(Uwada et al., Cellular Sig 2017 vol 35; Xie et al., Am J Physiol Gastroint Liver Physiol 2009 vol 296; Xie & Raufman, J Cancer Metastasis Treat 2016 vol 2; Tang et al., J Biol Chem 2002 vol 277; Ukegawa et al., J Cancer Res Clin Oncol 2003 vol 129). 

Moreover, a previous report indicated that carbachol at 1mM was not able to modify EGFR expression in a transactivation model performed in keratinocytes. Only the addition of exogenous EGF induced EGFR internalization in this model. (Ockenga et al., Int J Mol Sci 2014 vol 15). Our results are completely different since we observed a down-regulation in EGFR protein expression in tumor cells after the treatment (48h) with the combination of paclitaxel plus carbachol or APE at low doses.

Even though that the activation of EGFR is a main factor in the regulation of tumor growth; several authors consider that an up-regulation of EGFR expression is a bad prognosis marker and an indicator of chemoresistance in triple negative breast cancer (TNBC) patients or cell lines:

a. Zhang et al., Biomed Res Int. 2015;2015:357485 pointed to the distinct expression of EGFR and the prevalence of BRCA1 mutation indicating that EGFR, and BRCA1 might be unique biomarkers for targeted therapy and prognosis in TNBC.

b. Turner et al., Sci Rep. 2020 Jan 30;10(1):1493 concluded that EGFR is highly expressed in basal-like PDXs, cell lines, and patients, and high expression of this gene reduces metastasis-free survival, suggesting that targeting of this protein holds promise for potential clinical success in TNBC.

c. Islam et al., in J Cell Physiol. 2020 Jan 20. doi: 10.1002/jcp.29466 indicate that, the chemotolerance mechanism of TNBC has not yet been studied in detail. Infrequent messenger RNA expression, gene amplification (10-32.5%), and mutation (1%) of EGFR were seen in the TNBC samples irrespective of therapy, suggesting the importance of EGFR protein stabilization in this tumor. Thus, this study showed that reduced nuclear expression of Y654-p-β-catenin in NACT samples due to down-regulation of EGFR protein through promoter hypomethylation-mediated up-regulation of SH3GL2, resulting in low proliferation index/CD44 prevalence with better prognosis of the NACT patients, might have an important role in the chemotolerance of TNBC.

Although we cannot assure that EGFR present in tumor cells are fully functional, we measured a decrement in EGFR and ABCG2 expression due to the same treatment. Liu et al (Brain Res. 2015. 1611:93-100) reported an important link between EGFR expression and ABCG2 in U251 cells. They documented that leucine-rich repeats and immunoglobulin-like domains 1(LRIG1) can improve the chemosensitivity of these tumor cells. The presence of LRIG1 can reverse MDR by inhibiting epidermal growth factor receptor (EGFR) and secondary inhibiting ATP-binding cassette, sub-family B member 1(ABCB1) and ATP-binding cassette, sub-family G (WHITE), member 2 (ABCG2). A similar mechanism can be addressed in our model since the treatment with paclitaxel plus APE reduced the expression of both mentioned proteins.

However, this is only an assumption given the impossibility of conducting new experiments in the context of the current COVID-19 pandemia and the strict quarantine decreed by the Argentine government and the lack of an activity restart date. 

Paragraphs were modified for a better understanding of the results:

In Discussion and Conclusions:

• We removed ”possible”. Discussion and conclusions. Line 526.

• We added “to”. Discussion and conclusions. Line 527.

• We removed “of”. Discussion and conclusions. Line 527.

• We removed “, and whose”. Discussion and conclusions. Line 527.

• We added “. Its expression and/or activity”. Discussion and conclusions. Line 528.

• We removed “is related to the increment in proliferation and invasion, and”. Discussion and conclusions. Lines 528-529.

• We added “Several authors consider that the sole up-regulation of EGFR expression is a bad prognosis marker and an indicator of chemoresistance in TNBC patients or cell lines [39-41].”. Discussion and conclusions. Lines 529-531.

• We added “Although we only evaluated a down-regulation in the expression of EGFR due to drug treatment, more experiments could be useful to evaluate its activity in our system.”. Discussion and conclusions. Lines 536-538.

• We removed “Considering the link previously stated between ABC transporters´ expression and EGFR in glioblastomas, EGFR should be considered when designing a treatment for TN tumors in breast cancer patients.”. Discussion and conclusions. Lines 538-540.

In References:

• We removed the reference “Xu R, Shang C, Zhao J, Han Y, Liu J, Chen K, et al. Activation of M3 muscarinic receptor by acetylcholine promotes non-small cell lung cancer cell proliferation and invasion via EGFR/PI3K/AKT pathway. Tumor Biol. 2015; 36:4091-4100.”. References. Lines 691-693.

• We added the reference “Cristofaro I, Alessandrini F, Spinello Z, Guerriero C, Fiore M et al. Cross Interaction between M2 Muscarinic Receptor and Notch1/EGFR Pathway in Human Glioblastoma Cancer Stem Cells: Effects on Cell Cycle Progression and Survival. Cells. 2020; doi: 10.3390/cells9030657.”. References. Lines 694-697.

• We added the reference “Li Zhang, Cheng Fang, Xianqun Xu, Anling Li, Qing Cai, Xinghua Long. Androgen receptor, EGFR, and BRCA1 as biomarkers in triple-negative breast cancer: a meta-analysis. Biomed Res Int. 2015; doi: 10.1155/2015/357485.”. References. Lines 698-700.

• We added the reference “Turner T, Alzubi M, Harrell J. Identification of synergistic drug combinations using breast cancer patient-derived xenografts. Sci Rep. 2020; doi: 10.1038/s41598-020-58438-0.”. References. Lines 701-703.

• We added the reference “Islam S, Dasgupta H, Basu M, Roy A, Alam N, Roychoudhury S et al Reduction of nuclear Y654‐p‐β‐catenin expression through SH3GL2‐meditated downregulation of EGFR in chemotolerance TNBC: Clinical and prognostic importance. Cell Physiol. 2020; DOI: 10.1002/jcp.29466.”. References. Lines 704-707.

2.- Additional titrating experiments are needed for APE and methoctramine. 

APE and methoctramine are not absolute muscarinic receptor M2-selective agonist and antagonist. With the concentrations used in the studies, they cannot distinguish which muscarinic receptor subtypes are involved in the effect of muscarinic receptor activation. The author has to either revise the manuscript to remove the description of specific involvement of the M2 receptor or perform additional experiments by titrating the dose of drugs used.

We want to add additional results confirming M2 receptor participation in the effect exerted by APE + PX by silencing M2 receptor subtype with specific si-RNA obtained in Western blot and viability assays. These results show that the action of the combination on cell viability was similarly decreased by silencing the expression of M2 receptor or by pretreating cells with methoctramine, confirming the participation of this receptor subtype. 

Paragraphs were modified for a better understanding of the results:

In Materials and Methods:

• We added the item “M2 muscarinic receptor silencing”. Materials and methods. Lines 133-155.

In Results:

• We added “In addition, we confirmed the participation of M2 receptor by pretreating cells with a specific M2-siRNA that prevented the action of APE plus PX on MDA-MB231 cell viability (Supplementary Fig 1). ”. Results. Lines 333-336.

In Discussion and Conclusions:

• We added “of the combination either”. Discussion and conclusions. Line 509.

• We removed “preferential”. Discussion and conclusions. Line 510.

• We added “or by transfecting cells with a specific M2-siRNA”. Discussion and conclusions. Lines 510-511.

In Acknowledgments:

• We added “Dr. Federico Penas for providing NC-siRNA,”. Acknowledgments. Lines 577-578.

In Supporting information:

• We added the item “S1 Fig. Muscarinic subtype 2 receptor expression silencing.”. Supporting information. Lines 726-737.

• We added “S1 Fig”.

3.- Scratch assay is not an invasion assay but a migration assay. Please add the details of how this assay was performed (controls etc.)

How was the wound healing assay performed? Was an inhibitor of cell proliferation included? If not, then the effect of treatment on wound healing may be secondary to its effect on cell proliferation.

In this migration protocol, after adherence, cells were deprived of FBS to synchronized them. Then, the wound in the cell monolayer was done with a 200 µl pipette tip, washed twice with PBS and fresh culture medium without FBS containing different drugs was added.

Starvation conditions and total time for these experiments allowed us to quantify only cell migration, negligible considering cell division, since MDA-MB231 cells doubling time is greater than the time employed in each assay (https://physics.cancer.gov/docs/bioresource/breast/NCI-PBCF-HTB26_MDA-MB-231_SOP-508.pdf). 

Paragraphs were modified for a better understanding of the results:

In Materials and Methods:

• We added “supplemented with 10% FBS.” Materials and methods. Line 199.

• We added “Then, cells were deprived of FBS to synchronized them.”. Materials and methods. Line 200.

• We added “to produce a wound”. Materials and methods. Line 201.

• We added “a”. Materials and methods. Line 201.

• We added “without FBS”. Materials and methods. Line 202.

• We added “The conditions selected, allowed us to quantify cell migration, considering negligible cell division, since MDA-MB231 cell doubling time is longer than total experimental lasting for each assay.”. Materials and methods. Lines 205-207.

4.- Tumor angiogenesis assays were set up inappropriately. Angiogenesis requires the tumor to be formed, which process may take a few weeks in this particular strain. Thus any changes in the vessel density observed within 6 days post injection may reflect the death of the injected cells. Please modify you conclusions appropriately

Although other protocols have been described to determine angiogenesis, the one presented here was specially design to study tumor-induced angiogenesis in vivo by our group in 1996 (Monte et al., Eur J Cancer. 1997 vol 33) . With our methodology the intradermical administration of tumor cells (in the optimal concentration obtained by titration tests) in both flanks of mice allows, after 5 or 6 days of inoculation to determine the number of vessels induced by tumor cells and to discard those induced only by inflammation after injection. Also blind vessels and loops, typical of malignant vascularization can be distinguished and counted in amplified photographs of inoculated sites. In our method, the interactions among tumor, endothelial and other stromal cells can be considered while in other methods it is not possible to consider them simultaneously. 

Our method has been accepted by several reviewers in numerous publications from our lab listed down, including PLOS one in 2013.

-Effect of low dose metronomic therapy on MCF-7 tumor cells growth and angiogenesis. Role of muscarinic acetylcholine receptors.Salem AR, et al., Int Immunopharmacol. 2020 84:106514. 

-Angiogenesis signaling in breast cancer models is induced by hexachlorobenzene and chlorpyrifos, pesticide ligands of the aryl hydrocarbon receptor. Zárate LV, et al. Toxicol Appl Pharmacol. 2020 15;401.

-Denatonium and Naringenin Promote SCA-9 Tumor Growth and Angiogenesis: Participation of Arginase.Dmytrenko G, et al. Nutr Cancer. 2017;69:780 

-Synthetic stigmasterol derivatives inhibit capillary tube formation, herpetic corneal neovascularization and tumor induced angiogenesis: Antiangiogenic stigmasterol derivatives.

Michelini FM, et al., Steroids. 2016;115:160.

-Hexachlorobenzene promotes angiogenesis in vivo, in a breast cancer model and neovasculogenesis in vitro, in the human microvascular endothelial cell line HMEC-1. Pontillo C, et al. Toxicol Lett. 2015 19;239.

-Autoantibodies against muscarinic receptors in breast cancer: their role in tumor angiogenesis. Lombardi MG, et al. PLoS One. 2013; 8:e57572. 

-A natural antiviral and immunomodulatory compound with antiangiogenic properties. Bueno CA, et al. Microvasc Res. 2012, 84(3):235.

-Role of non-neuronal cholinergic system in breast cancer progression. Español A et al. Life Sci. 2007 ;80: 24.

-Activation of muscarinic cholinergic receptors induces MCF-7 cells proliferation and angiogenesis by stimulating nitric oxide synthase activity. Fiszman GL, et al. Cancer Biol Ther. 2007 6:1106.

- Muscarinic receptors autoantibodies purified from mammary adenocarcinoma-bearing mice sera stimulate tumor progression. Fiszman G et al., Int Immunopharmacol . 2006 6:1323.

- Muscarinic receptors are involved in LMM3 tumor cells proliferation and angiogenesis.Rimmaudo LE, et al. Biochem Biophys Res Commun. 2005; 334:1359.

-Muscarinic receptors participation in angiogenic response induced by macrophages from mammary adenocarcinoma-bearing mice.de la Torre E,et al. Breast Cancer Res. 2005;7: R345-52.

-Different mechanisms lead to the angiogenic process induced by three adenocarcinoma cell lines.

Davel LE, Rimmaudo L, Español A, de la Torre E, Jasnis MA, Ribeiro ML, Gotoh T, de Lustig ES, Sales ME.Angiogenesis. 2004;7(1):45.

-Arginine metabolic pathways involved in the modulation of tumor-induced angiogenesis by macrophages.Davel LE, et al., FEBS Lett. 2002;532:216-20. 

-Nitric oxide synthase-cyclooxygenase interactions are involved in tumor cell angiogenesis and migration.Davel L, et al., J Biol Regul Homeost Agents. 2002 16:181-9.

Paragraphs were modified for a better understanding of the results:

In Materials and Methods:

• We added “These animals are frequently used in xenogeneic assays with human cells. Mice.” Materials and methods. Lines 212-213.

• We removed “present a deprived immune system that allows inoculation and proliferation of human cells. Animals”. Materials and methods. Lines 213-214.

• We added “Briefly”. Materials and methods. Line 223.

• We removed “Concisely”. Materials and methods. Line 223.

• We added “and adjusted to”. Materials and methods. Line 224.

• We removed “form”. Materials and methods. Line 224.

• We added “of cell suspension”. Materials and methods. Line 225.

• We removed “containing 3x105 cells”. Materials and methods. Line 225.

• We added “into the area surrounding the third mammary fat pad in”. Materials and methods. Line 226.

• We removed “female NUDE”. Materials and methods. Line 227.

In Discussion and Conclusions:

• We added “The treatment of tumor cells with the combination of PX plus muscarinic agonists produced a decrement in VEGF-A levels in vitro. We cannot discard that the anti-angiogenic actions observed in vivo could also be due to a decrement in the number of tumor cells, or to a down-regulation of other angiogenic factors produced by tumor cells and/or by stromal cells. More experiments are needed to clarify these aspects in our model.”. Discussion and conclusions. Lines 560-565.

5.- The studies appear to be a replicate of the similar studies performed by the same group as noted in ref 16 and 17, except for using different cell lines. What are the new information provided in this manuscript?

Here, we demonstrated for the first time the efficacy of metronomic therapy targeting M2 receptors, with a low dose combination of paclitaxel plus APE. This combination produced a decrement in cell viability concomitantly with a down regulation in ABCG2 and EGFR expression. These results may be encouraging in relation to a decrement in resistance during the treatment of triple negative breast cancer. Moreover we proved for the first time the efficacy of the in vivo administration of the later combination to exert anti-angiogenic actions in a human triple negative breast cancer model. We also extended our observation to the usage of another traditional chemotherapeutic agent as doxorubicin at low doses in the combination. Furthermore, another type of triple negative cell line (MDA-MB468) was successfully treated with the combination obtaining similar effects on cell viability.

6.- Finally, the manuscript has numerous typos and grammatical errors and would benefit significantly from scientific writer editing from a native English speaker.

Corrections of typing and grammatical errors was performed by a native English speaker.

Paragraphs were modified for a better understanding of the results:

In Abstract:

• We removed “(M2)”. Abstract. Line 38.

• We removed “M2”. Abstract. Line 43.

• We added ”subtype 2 muscarinic”. Abstract. Line 43.

In Introduction:

• We removed “yet”. Introduction. Line 47.

• We added ”still”. Introduction. Line 47.

• We added ”acetylcholine”. Introduction. Line 67.

• We removed “acethylcholyne”. Introduction. Line 68.

• We added ” subtype 2 muscarinic (M2)”. Introduction. Line 82.

• We removed “The”. Introduction. Line 85.

• We removed “of low doses”. Introduction. Line 88.

• We removed “and”. Introduction. Line 89.

• We added ”with”. Introduction. Line 89.

• We added ”both at low doses”. Introduction. Line 89.

• We removed “non selective”. Introduction. Line 92.

• We added ”non-selective”. Introduction. Line 93.

• We removed “a”. Introduction. Line 93.

• We added ”an”. Introduction. Line 93.

• We added ”respectively”. Introduction. Line 93.

In Materials and Methods:

• We removed “by”. Materials and methods. Line106.

• We added “from”. Materials and methods. Line106.

• We removed “removed”. Materials and methods. Line110.

• We added “detached ”. Materials and methods. Line110.

• We removed “seen”. Materials and methods. Line128.

• We added “visualized”. Materials and methods. Line128.

• We removed “the”. Materials and methods. Line128.

• We removed “of the bands”. Materials and methods. Line129.

• We added “expressed”. Materials and methods. Line 129.

• We removed” displayed”. Materials and methods. Line129.

• We removed “the”. Materials and methods. Line129.

• We removed “arecaidine propargyl ester (”. Materials and methods. Line166.

• We removed “)”. Materials and methods. Line166.

• We added “during”. Materials and methods. Line 183.

• We removed “for”. Materials and methods. Line 183.

• We added “/each”. Materials and methods. Line 183.

• We removed “with the treatments”. Materials and methods. Line 183.

• We added “three”. Materials and methods. Line 230.

• We removed “3”. Materials and methods. Line 230.

In Results:

• We removed “Because of”. Results. Line 283.

• We added “Due to”. Results. Line 283.

• We added “produced a decrement in cell”. Results. Line 287.

• We removed “decreased”. Results. Line 287.

• We added “, when it was added to”. Results. Line 288.

• We removed “in”. Results. Line 288.

• We removed “method of”. Results. Line 309.

• We added “method”. Results. Line 309.

• We removed “indicate”. Results. Line 310.

• We added “indicated”. Results. Line 310.

• We added “This”. Results. Line 325.

• We removed “The latter”. Results. Line 325

• We added “Considering”. Results. Line 328.

• We removed “Taking into account”. Results. Line 328

• We added “a drug”. Results. Line 350.

• We removed “confirm”. Results. Line 380.

• We added “confirmed”. Results. Line 380.

• We removed “for”. Results. Line 381.

• We added “added in”. Results. Line 381.

• We added “could contribute”. Results. Line 401.

• We removed “contributes”. Results. Line 401.

• We added “exert”. Results. Line 443.

• We removed “exerted”. Results. Line 443.

In Discussion and conclusions:

• We added “long-term”. Discussion and conclusions. Line 481.

• We removed “long term”. Discussion and conclusions. Line 481.

• We added “of”. Discussion and conclusions. Line 482.

• We removed “either”. Discussion and conclusions. Line 482.

• We removed “breast”. Discussion and conclusions. Line 483.

• We added “of breast tumors, in particular TN”. Discussion and conclusions. Lines 486-487.

• We removed “breast tumors”. Discussion and conclusions. Line 487.

• We added “but it also produced similar actions on normal”. Discussion and conclusions. Line 491.

• We removed “also that it affects simultaneously non-tumorigenic”. Discussion and conclusions. Lines 491-492.

• We added “an undesirable”. Discussion and conclusions. Line 492.

• We removed “side”. Discussion and conclusions. Line 492.

• We added “the administration”. Discussion and conclusions. Line 503.

• We added “than MDA-MB231”. Discussion and conclusions. Line 505.

• We added “preventing”. Discussion and conclusions. Line 509.

• We removed “reversing”. Discussion and conclusions. Line 509.

• We removed “on one hand”. Discussion and conclusions. Line 542.

• We removed “On the other hand”. Discussion and conclusions. Lines 543-543.

• We added “In addition”. Discussion and conclusions. Line 545.

• We removed “in MDA-MB231 cells”. Discussion and conclusions. Line 546.

• We removed “In light of this background, it is expected that the combination of drugs designed in this work reduces proliferation and angiogenesis as observed in our results obtained in vitro and in vivo respectively.”. Discussion and conclusions. Lines 548-550.

• We added “and/or migration are the”. Discussion and conclusions. Line 551.

• We removed “is one of the”. Discussion and conclusions. Line 551.

• We added “they are”. Discussion and conclusions. Line 552.

• We removed “it is”. Discussion and conclusions. Line 552.

• We added “a”. Discussion and conclusions. Line 568.

• We added “of”. Discussion and conclusions. Line 571.

• We removed “in drug”. Discussion and conclusions. Line 571.

Upon re-submitting your revised manuscript, please upload your new study´s minimal underlying data set.

We attached a new excel file named “S4 minimal data set (S1 Fig)”.

PLOS ONE now requires that authors provide the original uncropped and unadjusted images underlying all blot or gel results reported.

We attached a new PDF file named “S3 raw images (S1 Fig)”.

---

## [Decision Letter · Decision Letter 2]

19 Aug 2020

The metronomic combination of paclitaxel with cholinergic agonists inhibits triple negative breast tumor progression. Participation of M2 receptor subtype

PONE-D-19-32678R2

Dear Dr. Español,

We’re pleased to inform you that your manuscript has been judged scientifically suitable for publication and will be formally accepted for publication once it meets all outstanding technical requirements.

Kind regards,

Irina V. Lebedeva, Ph.D.

Academic Editor

PLOS ONE

Additional Editor Comments (optional):

Reviewers' comments:

Reviewer's Responses to Questions

**Comments to the Author**

1. If the authors have adequately addressed your comments raised in a previous round of review and you feel that this manuscript is now acceptable for publication, you may indicate that here to bypass the “Comments to the Author” section, enter your conflict of interest statement in the “Confidential to Editor” section, and submit your "Accept" recommendation.

Reviewer #1: All comments have been addressed

Reviewer #2: All comments have been addressed

Reviewer #3: All comments have been addressed

Reviewer #4: All comments have been addressed

2. Is the manuscript technically sound, and do the data support the conclusions?

Reviewer #1: (No Response)

Reviewer #2: Yes

Reviewer #3: Yes

Reviewer #4: Yes

3. Has the statistical analysis been performed appropriately and rigorously? 

Reviewer #1: (No Response)

Reviewer #2: Yes

Reviewer #3: Yes

Reviewer #4: Yes

4. Have the authors made all data underlying the findings in their manuscript fully available?

Reviewer #1: (No Response)

Reviewer #2: Yes

Reviewer #3: Yes

Reviewer #4: Yes

5. Is the manuscript presented in an intelligible fashion and written in standard English?

Reviewer #1: (No Response)

Reviewer #2: Yes

Reviewer #3: Yes

Reviewer #4: Yes

6. Review Comments to the Author

Reviewer #1: (No Response)

Reviewer #2: All comments are addressed properly in the manuscript entitled "The metronomic combination of paclitaxel with cholinergic agonists inhibits triplenegative breast tumor progression. Participation of M2 receptor subtype"

Reviewer #3: The author has largely addressed the reviewers' comments. The M2 silencing studies provided strong evidence for the involvement of this receptor in enhancing the activity of paclitaxel.

Reviewer #4: The authors have addressed all comments appropriately, and no significant concerns remain. Would recommend publication.

7. PLOS authors have the option to publish the peer review history of their article (what does this mean?). If published, this will include your full peer review and any attached files.

Reviewer #1: No

Reviewer #2: No

Reviewer #3: No

Reviewer #4: No

---

## [Editor Report · Acceptance letter]

25 Aug 2020

PONE-D-19-32678R2 

The metronomic combination of paclitaxel with cholinergic agonists inhibits triple negative breast tumor progression. Participation of M2 receptor subtype. 

Dear Dr. Español:

I'm pleased to inform you that your manuscript has been deemed suitable for publication in PLOS ONE. Congratulations! Your manuscript is now with our production department. 

Kind regards, 

on behalf of

Dr. Irina V. Lebedeva 

Academic Editor

PLOS ONE